# Children exhibit a developmental advantage in the offline processing of a learned motor sequence
Anke Van Roy, Geneviève Albouy, Ryan D. Burns & Bradley R. King ✉

Changes in specific behaviors across the lifespan are frequently reported as an inverted-U trajectory. That is, young adults exhibit optimal performance, children are conceptualized as developing systems progressing towards this ideal state, and older adulthood is characterized by performance decrements. However, not all behaviors follow this trajectory, as there are instances in which children outperform young adults. Here, we acquired data from 7–35 and >55 year-old participants and assessed potential developmental advantages in motor sequence learning and memory consolidation. Results revealed no credible evidence for differences in initial learning dynamics among age groups, but 7- to 12-year-old children exhibited smaller sequence-specific learning relative to adolescents, young adults and older adults. Interestingly, children demonstrated the greatest performance gains across the 5 h and 24 h offline periods, reflecting enhanced motor memory consolidation. These results suggest that children exhibit an advantage in the offline processing of recently learned motor sequences.

The ability to learn novel movements is vital to the everyday functioning of individuals across the human lifespan (e.g., from a child learning to ride a bike to an older individual learning to operate a touchscreen). One of the dominant motor learning paradigms in the field is motor sequence learning (MSL), whereby participants acquire a novel series of interrelated actions (i.e., a motor sequence). MSL has been extensively studied with multiple task variants, including the serial reaction time task (SRTT; i.e., participants respond to visual cues presented in a repeating order) and the finger tapping task (i.e., participants repeatedly reproduce a known/shown sequence of finger movements in a self-initiated manner). There has been extensive previous research that has employed these MSL task variants in healthy young adults (~18–35 years), facilitating the development of a framework that characterizes the time course of sequence learning in this age group (see refs. 1–4 for reviews). Typically, a fast online (i.e., during repeated practice) acquisition phase with substantial performance improvements occurs during an initial training session. This is followed by a slow offline (i.e., in the absence of active task practice) consolidation phase that spans hours during which the acquired memory trace is stabilized, strengthened and reorganized into a more robust form.

Although previous research has examined MSL in school-aged children (i.e., between 5 and 12 years), the results are quite heterogenous. While some studies have reported comparable learning within the initial practice session in children and adults[5–8], others have shown that children exhibit worse[9–13] or better learning[10,14–17]. There is a general consensus, however, that children exhibit impaired consolidation of a recently acquired motor sequence over offline periods that include sleep [e.g., refs. 13,15,18, but see refs. 8,19 for specific examples of sleep-related benefits]. Specifically, whereas post-learning sleep is known to boost offline consolidation processes in young adults [see ref. 3 for a review], such a beneficial effect is absent in children. Interestingly, there is some evidence from sequential finger tapping task variants suggesting that this degraded consolidation over sleep may be the by-product of enhanced or accelerated consolidation over the wake epochs shortly following initial learning[9,14,20]. Specifically, children showed a rapid performance stabilization and enhancement 15 min[9] and 1 h[14] after learning, which were not observed in young adults.

Interestingly, this enhanced offline processing does not appear to be limited to the timescale of hours following initial learning but is also observed on shorter timescales. Specifically, Du et al.[5] employed a stepping version of an SRTT and found that young children (i.e., 6-year-olds) showed larger performance improvements over the short rest intervals in between blocks of practice within a session as compared to older children (i.e., 10-year-olds) and young adults. These results collectively suggest that young children may exhibit a developmental advantage with respect to the micro- (i.e., timescale of seconds between practice blocks) and macro- (i.e., timescale of hours between practice sessions) offline processing of recently practiced motor sequences during wake intervals. This childhood advantage

Department of Health and Kinesiology, College of Health, University of Utah, Salt Lake City, UT 84112, USA. ✉e-mail: Bradley.ross.king@utah.edu

could then be added to the list of examples in which children outperform adults in specific learning and memory behaviors (for review see[21]).

At the other end of the lifespan, there is considerable evidence that older adults (>~60 years) exhibit comparable initial learning of a novel motor sequence as young adults[22–26]. And, whereas children appear to have a developmental advantage in the offline processing of recently acquired sequences, older adults are known to exhibit deficits. Specifically, the magnitude of offline gains over macro-offline intervals that include sleep and/or wake is significantly less than those observed in younger adults[22,24–28].

Although the existing literature suggests that the offline processing of recently acquired movement sequences appears to be superior in children and then declines throughout adulthood, systematic examinations into changes across the human lifespan are relatively limited (but see refs. 10, 11, 16,29 for examples in the domain of implicit sequence learning). This study thus aimed to provide an exhaustive characterization of motor learning and memory consolidation behaviors across the lifespan (i.e., in groups of children, adolescents, young adults and older adults). Consistent with previous research[5–8], we hypothesized that the initial learning of a motor sequence (i.e., within a single training session) would be comparable across the 4 age groups. We expected children to exhibit enhanced micro- and 5-h macro-offline consolidation in comparison to young adults, indicative of a developmental advantage in the offline processing of a recently acquired motor sequence across periods of wakefulness. This developmental advantage in macro-offline consolidation was expected to be absent in the 24-h retest, as previous research has indicated impaired sleep-facilitated consolidation in children[8,13,15,18]. Last, and consistent with earlier studies, older adults were expected to demonstrate intact initial learning but impaired macro-offline consolidation across both 5- and 24-h intervals when compared to the young adults.

## Methods

The project consisted of two experimental protocols. Specifically, Experiments 1 and 2 examined the effect of age on the initial acquisition of a movement sequence and the time course of motor memory consolidation, respectively. Data collection and analysis plans were pre-registered in March and August 2021 via the Open Science Framework and can be accessed at https://doi.org/10.17605/OSF.IO/WBK9H and https://doi.org/10.17605/OSF.IO/ZMJ75. Any additional analyses that were not included in the pre-registrations are labeled in this text as exploratory. Note that 4 (out of 130) and 12 (out of 108) datasets were acquired prior to the pre-registrations of Experiments 1 and 2, respectively. This was done to verify that keypresses and their timing were logged adequately, and that saved data were complete. The processed data used for the results presented in this text as well as the

raw data are publicly available on Zenodo (https://doi.org/10.5281/zenodo.8274118).

## Participants

Healthy volunteers between 7–35 and 55–75 years old of all genders were recruited by advertisements on relevant websites and research databases. Note that as our experiments were conducted through a web-based data collection platform, recruitment was not constrained to a specific geographical region. Individuals interested in participation were sent an online screening questionnaire to assess eligibility, which was subsequently reviewed by a member of the research team. Exclusion criteria were: (1) reported history of medical, neurological, psychological or psychiatric conditions, (2) use of psychoactive or sleep-influencing medications, (3) indications of abnormal or irregular sleep, (4) mobility limitations of the fingers or hands, or (5) considered a professional typist or prior extensive training on a musical instrument requiring dexterous finger movements (e.g., piano, guitar). All experimental procedures were approved by the University of Utah Ethics Committee (IRB_00136894). Adult participants and parents of underage participants gave informed consent and participants below 18 years of age provided informed assent. Participants received an electronic gift card with a value of $15 or $30 (Experiments 1 and 2, respectively) as compensation for their participation.

A convenience sample of 224 individuals met the inclusion criteria and thus initiated participation in an experimental protocol. They were divided into the following four age groups: children (operationally defined as 7–12 years old), adolescents (13–17 years old), young adults (18-35 years old) and older adults (≥55 years old). Pre-registered analyses compared differences among these age *groups*. Yet, to provide more fine-grained analyses of age-related trajectories of motor learning and memory consolidation behaviors, we also conducted exploratory analyses with age as a continuous variable (see below for details).

Sample size computations were conducted *a priori* and with the software G*Power[30]. For the assessment of age-group differences in initial motor sequence learning (i.e., Experiment 1), and to detect an effect size of $f = 0.3$ (based on the comparison of initial learning between 9-year-olds and young adults in Adi-Japha et al.[9]), with an alpha of 0.05 and power of 0.80, the desired sample size was 128 subjects (32 per group). Of the participants that initiated the motor learning protocol, data from 7 participants were excluded from analyses due to a failure to comply to experimental instructions (e.g., repeatedly pressing the same key; $n = 3$ children), missing data because of software issues ($n = 1$ young adult), or a failure to correctly perform the motor task (i.e., statistical outliers (>3 SD from group mean) on sequence accuracy as defined in the preregistration; $n = 1$ child, 1 young adult, 1 older adult). Excluded participants were replaced and thus the final sample size for analyses consisted of 130 participants (see Table 1 for participant demographics).

For the assessment of age-group differences in macro-offline consolidation processes (i.e., Experiment 2), the detection of an effect size of $f = 0.20$ (slightly more conservative than Adi-Japha et al.[9]), with an alpha of 0.05, a power of 0.80 and a correlation among repeated measurements of 0.25, this experiment required a sample size of 108 (27 participants per group). Of those participants that initiated the motor learning protocol, 21 individuals were excluded from data analysis. Specifically, 17 participants failed to comply to experimental instructions (i.e., 4 children repeatedly pressed the same key and 5 children, 2 young adults and 6 older adults either did not complete the protocol or did not adhere to the specific schedule of the experimental sessions). One older adult was excluded due to inaccurate data because of software issues. And, based on our preregistration, additional exclusions were due to a lack of performance improvements across training ($n = 1$ young adult, 1 older adult) and an inability to correctly perform the motor task (statistical outliers on sequence accuracy; $n = 1$ young adult). Similar to above, excluded participants were replaced and thus the final sample for analyses consisted of 108 participants (see Table 2 for participant details). Data from the third session of one young adult and the post-learning random data of one adolescent were missing; thus, these

**Table 1 | Participant demographics, and sleep and vigilance scores for each age group in Experiment 1**

| Variable | Children | Adolescents | Young adults | Older adults | p-value |
|---|---|---|---|---|---|
| n | 33 | 33 | 32 | 32 | / |
| Female (n) | 17 | 12 | 23 | 24 | 0.004* |
| Age (years) | 10.2 (1.6) | 15.3 (1.3) | 26.3 (5.0) | 64.4 (5.2) | / |
| M/E Preference | 2.9 (0.9) | 3.3 (0.8) | 2.9 (1.0) | 2.3 (0.8) | <0.001* |
| Time of testing (SD in min) | 12:47 (172) | 12:29 (133) | 14:52 (175) | 12:05 (169) | <0.001* |
| Sleep quality | 5.1 (0.8) | 5.0 (0.7) | 4.8 (0.6) | 4.8 (0.8) | 0.12 |
| Sleep duration (hours) | 9.8 (0.9) | 9.1 (1.4) | 8.5 (1.0) | 7.7 (1.1) | <0.001* |
| SSS score | 2.0 (0.9) | 2.0 (0.8) | 2.0 (0.7) | 1.5 (0.7) | 0.031* |

Numbers represent the mean, with standard deviation in parentheses. Gender was determined by asking what gender the participant identified with the most. No participant reported to be non-binary. Morningness/Eveningness (M/E) Preference was added as an exploratory variable to assess age-related differences in circadian preferences and was defined on a 5-point Likert scale (i.e., 1 = extreme morning person, 5 = extreme evening person). The average time of testing is specified in 24-h time notation, with the SD in minutes. Sleep quality is defined on a 6-point Likert scale (i.e., 1 = very bad, 6 = very good). SSS = Stanford Sleepiness Scale[33], with higher numbers indicative of increased sleepiness. The p-values resulted from one-way ANOVAs assessing group differences. See Supplementary Table 1 for full statistical information.

**Table 2 | Participant demographics, and sleep and vigilance scores for each age group in Experiment 2**

| Variable | | Children | Adolescents | Young adults | Older adults | *p*-value |
|---|---|---|---|---|---|---|
| *n* | | 27 | 27 | 27 | 27 | / |
| Female *(n)* | | 16 | 12 | 21 | 18 | 0.08 |
| Age (years) | | 10.6 (1.6) | 15.4 (1.4) | 26.6 (4.4) | 63.0 (4.4) | / |
| M/E Preference | | 3.0 (0.8) | 3.2 (0.9) | 3.2 (0.8) | 2.4 (0.9) | 0.002* |
| Session 1 | Time of testing | 11:34 (97) | 11:34 (96) | 11:39 (82) | 11:30 (92) | 0.99 |
| | Sleep quality | 4.9 (0.7) | 5.1 (0.6) | 5.0 (0.6) | 4.7 (0.8) | 0.19 |
| | Sleep duration (hrs.) | 9.7 (1.1) | 9.4 (1.2) | 8.3 (1.1) | 7.7 (1.3) | <0.001* |
| | SSS score | 2.2 (0.9) | 2.1 (0.8) | 2.0 (0.7) | 1.5 (0.7) | 0.012* |
| | PVT score (ms) | 360.9 (81.5) | 341.9 (102.3) | 320.9 (62.0) | 327.5 (75.7) | 0.31 |
| Session 2 | Time of testing | 16:42 (99) | 16:56 (112) | 16:43 (89) | 16:29 (96) | 0.81 |
| | Offline period (hrs.) | 5.1 (0.8) | 5.4 (0.9) | 5.1 (1.1) | 5.0 (0.7) | 0.43 |
| | SSS score | 2.0 (1.0) | 1.8 (0.8) | 2.0 (0.7) | 1.7 (0.7) | 0.36 |
| | PVT score (ms) | 386.0 (88.4) | 318.5 (61.0) | 320.3 (69.1) | 323.6 (68.0) | 0.002* |
| Session 3 | Time of testing | 11:57 (148) | 12:25 (156) | 11:30 (97) | 11:24 (109) | 0.31 |
| | Offline period (hrs.) | 24.4 (1.8) | 24.8 (1.9) | 23.8 (1.4) | 23.9 (1.4) | 0.095 |
| | Sleep quality | 5.0 (0.8) | 4.7 (0.8) | 4.7 (0.7) | 4.6 (0.7) | 0.38 |
| | Sleep duration (hrs.) | 10.3 (0.9) | 9.3 (1.5) | 8.0 (1.3) | 7.6 (1.0) | <0.001* |
| | SSS score | 1.9 (0.7) | 2.3 (1.3) | 2.3 (0.8) | 1.5 (0.6) | 0.003* |
| | PVT score (ms) | 378.7 (97.3) | 325.8 (74.6) | 308.0 (54.4) | 306.3 (33.6) | <0.001* |

Numbers represent the mean, with standard deviation in parentheses. Gender was determined by asking what gender the participant identified with the most. No participant reported to be non-binary. Morningness/Eveningness (M/E) Preference was added as an exploratory variable to assess age-related differences in circadian preferences and is defined on a 5-point Likert scale (i.e., 1 = extreme morning person, 5 = extreme evening person). The average time of testing is specified in 24-h time notation, with the SD in minutes. The offline periods represent the time periods between session 1 and sessions 2 and 3, respectively. Sleep quality is defined on a 6-point Likert scale (i.e., 1 = very bad, 6 = very good). SSS = Stanford Sleepiness Scale[33], with higher numbers indicative of increased sleepiness. The PVT score indicates the average simple RT across the 30 trials. The *p*-values resulted from one-way ANOVAs assessing group differences. See Supplementary Discussion 1 for full statistical information.

individuals were excluded from the contrasts involving these specific task runs only.

Note that the procedures of Experiment 1 (assessing initial motor sequence learning) were identical to the first session of Experiment 2 (assessing macro-offline consolidation). Accordingly, data from a subset of early participants from Experiment 2 (*n* = 42; 16 children, 19 adolescents, 4 young adults, 3 older adults) were included in the analyses of Experiment 1. Thus, these participants were included in analyses for both experiments.

**General experimental procedures**

After verification of eligibility, participants were contacted via email with the necessary information to continue participation. This email contained a link to a video created by our study team that provided detailed instructions on how to complete our motor learning task (details on the task are provided below). Participants were also provided links to our series of tasks created on the web-based data collection platform PsyToolkit[31,32]. As our experiments were conducted online, participants were free to choose which day(s) to complete experiments but were instructed to complete the protocols on days in which they were well-rested and could adhere to the schedule of the sessions. The precise times-of-day that the experiments were completed were constrained and followed experiment-specific schedules provided below. Participants were instructed to receive a good night of sleep and to refrain from alcohol the night before and throughout the experiment. Compliance to these instructions was verified through a brief questionnaire prior to completion of the motor task. Lastly, parents of the child participants were asked to be present when their child completed the experiments.

For both experiments and at the beginning of each experimental session, a brief questionnaire was completed in which participants reported their sleep patterns in the previous 24 h and their subjective feelings of alertness (Stanford Sleepiness Scale[33]). Additionally, and for Experiment 2 only, participants completed the Psychomotor Vigilance Task (PVT[34]) to provide an objective measure of vigilance at the time of testing. The PVT

required participants to fixate on the middle of the computer screen and press the space bar as fast as possible when a visual stimulus appeared after a varying delay interval. Response time (i.e., time between appearance of stimulus and keypress) was logged and used to assess vigilance (i.e., higher RTs indicative of lower vigilance). We administered a shortened version of the PVT[35,36] that consisted of 35 trials, of which the first 5 were discarded as participants became familiar with the task.

**Serial Reaction Time Task (SRTT)**

All participants performed an adapted version of the explicit Serial Reaction Time Task (SRTT) similar to our previous research[37,38], which was coded in the online data acquisition platform PsyToolkit[31,32]. The SRTT was chosen for this research, in part, because of the online data acquisition protocol. It was assumed that participants, and young children in particular, could more easily comprehend and follow the instructions for the SRTT variant (i.e., press the key that spatially corresponds to the visual stimulus). This in contrast to the sequential finger tapping task, where fingers are assigned numerical values and participants are then instructed to perform the sequence of finger movements that corresponds to the explicitly provided sequence of numbers.

The SRTT employed in this research consisted of an 8-choice reaction time task in which participants were instructed to react to visual cues shown on a screen (see Fig. 1a). Eight squares that spatially corresponded to the eight fingers used to perform the task (i.e., no thumbs) and the eight keys on the keyboard were presented on the screen. The color of the squares alternated between red and green, indicating rest and practice blocks, respectively. During practice, a visual stimulus (i.e., a butterfly) appeared consecutively in one of the squares and participants responded to the visual cues by pressing the corresponding key with the corresponding finger as fast and accurately as possible. The task was presented as a game, designed to increase motivation in child participants, in which participants of all four age groups were instructed to move quickly to catch as many butterflies as

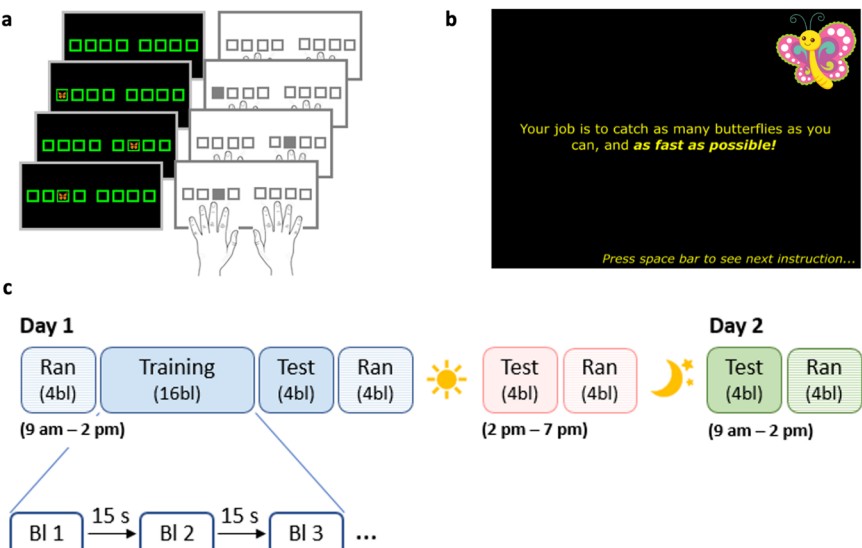

**Fig. 1 | Experimental procedures. a** Serial reaction time task (SRTT). A stimulus appeared in 1 of 8 spatial locations and the participant responded with the corresponding key/finger as fast as possible. Stimuli appeared in either a pseudorandom or sequential (i.e., 4-7-3-8-6-2-5-1) manner. **b** To increase motivation and attention, the task was built around a story in which participants were asked to catch butterflies. **c** Design of Experiments 1 and 2. Experiment 1 consisted of the first session only (i.e., Day 1 in blue) and Experiment 2 of all three sessions displayed. Session 1 included four blocks (bl) of pseudorandom SRTT (Ran), followed by 20 blocks of sequential SRTT, which were divided into 16 blocks of training and 4 blocks of test after a 1-min rest, and four additional blocks of pseudorandom SRTT. In Experiment 2, session 2 and 3 were completed approximately 5 and 24 h, respectively, after session 1. Both sessions 2 and 3 consisted of 4 blocks of sequential SRTT and 4 blocks of pseudorandom SRTT. Note that the first session of Experiment 2 was constrained between 9 am and 2 pm to give ample time for the 5 h delayed retest. The participants in Experiment 1 were instructed to complete the session anytime between 9 am and 7 pm. Butterfly images in **a** and **b** were taken from https://openclipart.org and fall under the Creative Commons Zero 1.0 Public Domain License.

possible (see Fig. 1b). The visual cues succeeded each other in either a pseudorandom or a sequential order, depending on the experimental run (i.e., pseudorandom SRTT vs. sequential SRTT, respectively). In the sequential SRTT, participants were aware that the stimuli and keypresses followed a deterministic sequential pattern (i.e., 4-7-3-8-6-2-5-1, in which 1 through 8 are the left pinky to the right pinky fingers from left to right), but they were not given any information about the structure or length of the sequence. All participants completed the same 8-element sequence. During the pseudorandom SRTT runs, the visual stimuli (and thus corresponding key presses) appeared in an order that pseudo-randomly changed every 8 elements. Specifically, the stimulus appeared in each location once every eight elements, but the stimulus never appeared in the same location consecutively. The order of the stimuli thus changed every eight elements within a block, across blocks of practice and differed across participants. For both pseudorandom and sequential task variants, practice blocks contained 48 key presses (i.e., 6 repetitions of the 8-element sequence in the sequence condition) and were separated by 15-s rest periods.

Measures reflecting performance speed (i.e., response time between visual cue and keypress) and accuracy (i.e., correct or incorrect keypress) of the participants' motor responses were recorded for each trial and used to assess performance on the task. Per our pre-registrations, individual trials (i.e., responses) were excluded from analyses if the measured response time was greater than 3 standard deviations above or below the participant's mean response time for that block. For Experiments 1 and 2, an average of 1.78% (children: 2.03%, adolescents: 1.76%, young adults: 1.68%, older adults: 1.65%) and 1.69% (children: 2.0%, adolescents: 1.83%, young adults: 1.58%, older adults: 1.34%) of trials were excluded from analyses, respectively.

### Experiment 1 design
The first experiment consisted of a single experimental session in which participants performed both the random and sequential variants of the SRTT (see Day 1 [blue] of Fig. 1c). Participants were instructed to complete

this session between 9 am and 7 pm (see Table 1 in the main text and Supplementary Note 1, Supplementary Table 1 and Fig. S1 for details on time of testing). The experimental session started with one block of familiarization (pseudorandom; data not analyzed) followed by four blocks of the pseudorandom SRTT (pre-learning random), which afforded an assessment of general motor execution on the task prior to any sequence learning. Subsequently, participants completed 20 blocks of the sequential SRTT. These blocks were divided into two runs: a training run that consisted of 16 blocks and then a post-learning test run of 4 blocks. The post-learning test run was completed approximately one minute after the end of training, affording the assessment of end-of-training performance following the further dissipation of mental and physical fatigue[39]. Lastly, four additional blocks of the pseudorandom SRTT (post-learning random) were completed, allowing us to assess whether improvements in performance on the sequential SRTT could be attributed to learning of the movement sequence *per se* (i.e., sequence-specific learning) or simply a general improvement in movement speed due to task familiarization.

### Experiment 2 design
The second experiment consisted of three experimental sessions (see Fig. 1c). Participants were instructed to complete the first session between 9 am and 2 pm. This session was identical to what was described for Experiment 1. The second session took place approximately five hours after the first session, allowing the assessment of motor memory consolidation after an interval of post-learning wakefulness. This second session consisted of four blocks of sequential SRTT during which participants again performed the same specific sequence as in session 1, and four blocks of pseudorandom SRTT (post-test random). Finally, the third session was completed 24 h after the first session, allowing the examination of motor memory consolidation after a night of sleep. Similar to the second session, this third session included four blocks of sequential SRTT and four blocks of pseudorandom SRTT (post-test random). See Table 2 in the main text as well as Supplementary Note 1, Supplementary Table 2 and Fig. S2 for details

on actual time of testing. For the intervals between sessions, participants were instructed to refrain from napping, consuming alcohol or recreational drugs and to receive a good night of sleep.

### Data processing and statistical analyses

We first describe general data processing and statistical analysis information prior to providing details specific to each of the two experiments. All null hypothesis statistical testing described below was performed using IBM SPSS Statistics for Windows, version 28 (IBM, Armonk, NY, USA). All significance tests were two-sided, considered significant when $p < 0.05$ and performed for both normalized outcome measures (i.e., reaction time and accuracy). For the sake of completeness, we also highlight those effects when $0.05 < p < 0.1$ as non-significant trends. In the event of a violation of the sphericity assumption, Greenhouse-Geisser correction was applied. All significant main effects and interactions were followed up by pairwise comparisons using the Tukey test and one-way ANOVAs, respectively. Depending on the statistical test, eta-squared, partial eta-squared or Hedges' g are reported as effect size measures. The correspondence among reported $p$-values, test statistics and degrees of freedom reported in this main text was verified via statcheck[40].

In addition to the null hypothesis significance testing outlined above, the anovabf, ttestbf and regressionbf functions in RStudio version 1.3.1093 (Posit, PBC, Boston, Massachusetts, USA) were used to compute Bayes Factors (BF) for all effects presented in the main text. Note that this Bayesian analysis was *not* included in the pre-registrations. BFs reflect the likelihood that the observed data favor an alternative model (e.g., evidence for differences among groups, blocks, sessions, etc.) relative to the null model (e.g., evidence for no differences). For *t*-tests, the null model was the difference between means equal to zero. For mixed model ANOVAs, the null models were specified as the random effect of subjects. For one-way ANOVAs and simple linear regressions, the null models were only the intercept term. For multiple linear regressions that assessed age group differences in the relationship between micro- and macro-offline performance gains (see below for details), the reported BFs reflect the comparison of two models with and without the age group x micro-offline gain interaction term. Default parameters in the R functions were used, with the exception of "whichModels" that was set to "all" for mixed model ANOVAs and multiple regressions in order to compute the appropriate BFs. In our results, we report $BF_{10}$ values and adhere to the interpretations offered in Wagenmakers et al.[41]; larger values are indicative of greater likelihood that the observed data favor the alternative as compared to the null hypotheses.

To assess potential differences among age groups and experimental sessions in certain demographic/participant characteristics, one-way ANOVAs or chi-square tests (Experiment 1) and two-way ANOVAs (Experiment 2) were performed. These measures included gender, vigilance (both subjective and objective) at the time of testing, self-reported sleep quantity and quality of the night prior to the experimental session(s), and the time of day in which the motor task or session was completed. Any significant effects from these analyses were followed up by pairwise comparisons to determine which age groups or sessions differed from each other. Group means for these dependent measures are provided in Tables 1 and 2 and full output from the corresponding statistical analyses can be found in Supplementary Tables 1 and 2.

For the SRTT, the averaged response times for correct keypresses and the percentage of correct keypresses were computed for each block and task variant. Additionally, as previous research has indicated age differences in motor performance independent of sequence learning[6,7,22,23,42], baseline differences in performance were accounted for by normalizing both movement speed and accuracy of the sequential and post-learning random SRTT runs relative to the outcome variables of the pre-learning random run. More specifically, for each outcome measure and each block of the SRTT runs, performance was divided by the mean outcome measure across the four blocks of the pre-learning random run. Statistical analyses of these normalized measures are presented in the main text. For completeness, results based on non-normalized

performance measures are reported in Supplementary Notes 2–4, Supplementary Tables 3, 4, and Figs. S3–S8.

**Experiment 1—data analysis.** Sequential learning dynamics (i.e., block-to-block performance changes) were compared among age groups using a mixed ANOVA, with *group* (i.e., children, adolescents, young adults, older adults) as between-subject and *block* (i.e., 16 training blocks or 4 test blocks) as within-subject factors.

To assess whether observed improvements in performance across practice blocks reflected sequence-specific learning (as opposed to general skill learning as a result of familiarization with the motor task), a learning magnitude measure was computed for performance speed. Specifically, the difference between averaged normalized performance across the four blocks in the post-learning test run of the sequential SRTT and the averaged normalized performance in the post-learning random run was divided by the average normalized performance in the post-learning random run. This measure thus reflects the relative difference between the random and sequential task variants at the end of training, affording the assessment of sequence-specific learning. Larger values are indicative of greater sequence-specific learning. The effect of age *group* on learning magnitude was assessed with a one-way ANOVA. Supplementary Note 3 and Fig. S5 include corresponding results for when this learning magnitude measure was computed on non-normalized RT data. It is worth noting here that the findings were consistent across the two computations.

To provide a more fine-grained characterization of the age-related changes in sequence-specific learning magnitude, we conducted exploratory analyses with age as a continuous variable. Specifically, we tested five potential fit options (i.e., single exponential, double exponential, linear, quadratic, and power functions) to characterize the relationship between age and learning magnitude. As we did not acquire data from participants between 35 and 55 years of age, separate models were built to characterize age-related changes from 7 to 35 and 55 to 75 years of age. The model with the lowest Akaike Information Criterion value was deemed as the best fitting model and the corresponding results are presented.

Similar to previous research[5,43,44], we distinguished between micro-online (i.e., within blocks of task practice) and micro-offline (i.e., across rest epochs in between blocks of practice) performance improvements. Micro-online changes were computed as the difference between the averaged normalized RT for the correct keypresses of the first and last sequence repetitions within each block (i.e., first sequence block $n$ – last sequence block $n$). For micro-offline changes, the difference between the averaged normalized RT of the last sequence repetition of one block and the first sequence repetition of the subsequent block (i.e., last sequence block $n$ – first sequence block $n + 1$) was computed. These micro-online and micro-offline measures were analyzed with separate *group(4)* x *block(16)* or *group(4)* x *rest block (15)* mixed ANOVAs, respectively. Supplementary Note 3 and Fig. S6 include corresponding results when micro-online and -offline measures were computed with non-normalized RT data. Last, and following the same exploratory procedure as described above for the learning magnitude measure, we assessed age-related changes in these micro measures with age as a continuous variable.

**Experiment 2—data analysis.** As in Experiment 1, learning dynamics (i.e., block-to-block performance changes) across training and the post-learning test were analyzed using *group(4)* x *block(16 or 4)* mixed ANOVAs to assess potential group differences in initial encoding of the motor sequence within the first session (i.e., before the assessment of consolidation).

The main focus of Experiment 2 was the effect of age on motor memory consolidation across the 5 h and 24 h macro-offline intervals. To this end, and for both the sequential and random SRTT, a macro-offline change measure was computed for each outcome variable and offline interval. Specifically, sequential offline changes were calculated as the difference between the average normalized performance on the sequential SRTT run in the 5 h and 24 h retest sessions and the average

normalized performance in the post-learning test run of the first session. Similarly, offline changes for the random task variant were computed as the difference between the averaged normalized performance on the post-test random runs in the 5 h and 24 h retest sessions and the averaged normalized performance in the post-learning random run of the first session. These measures reflect the offline changes in sequential performance and general motor execution from the end of the initial training session to the two retest sessions, respectively. Mixed ANOVAs with *group (4)* as between-subject and *offline period* (i.e., 5-h and 24-h offline periods) as within-subject factors were conducted. Supplementary Note 4 and Figs. S7 and S8 include corresponding results when macro-offline measures were computed with non-normalized RT data. Consistent with the exploratory analyses described above for Experiment 1, age-related changes in the macro-offline performance gains were also assessed with age as a continuous variable.

Last, and similar to previous research[44,45], we assessed the relationship between offline changes in performance on the micro- and macro-offline timescales. Specifically, we performed exploratory multiple regressions with the micro-offline gains as independent and the 5 h or 24 h macro-offline gains as dependent variables. To examine how these relationships differed among age groups, we included age group (dummy coded) main effects as well as age group x micro-offline gains interaction variables in the regression models. The young adult group was selected as the reference group and thus the interaction variables compare the regression slopes of the other 3 age groups with those of the young adults.

### Reporting summary
Further information on research design is available in the Nature Portfolio Reporting Summary linked to this article.

## Results
### Experiment 1—participant characteristics, sleep and vigilance
Group means from participant characteristics, the time of when the experimental session was completed, self-reported sleep quality and duration of the night preceding the experimental session, and a subjective measure of vigilance are provided in Table 1. Results from the corresponding statistical analyses and a brief discussion can be found in Supplementary Note 1 and Supplementary Table 1.

### Experiment 1—dynamics of performance improvements
Normalized performance measures on the SRTT are depicted in Fig. 2a, b. Performance accuracy remained stable across training, as evidenced by no main effect of block ($F_{(7.011,883.403)} = 0.501$, $p = 0.835$, partial $\eta^2 = 0.004$, $BF_{10} = 3.153e^{-06}$), and did not significantly differ among age groups ($F_{(3,126)} = 0.322$, $p = 0.810$, partial $\eta^2 = 0.008$, $BF_{10} = 0.091$). Furthermore, no significant group x block interaction was found ($F_{(21.033,883.403)} = 0.715$, $p = 0.821$, partial $\eta^2 = 0.017$, $BF_{10} = 1.821e^{-05}$). Similarly, no block ($F_{(2.283,287.681)} = 1.523$, $p = 0.217$, partial $\eta^2 = 0.012$, $BF_{10} = 0.060$), group ($F_{(3,126)} = 1.796$, $p = 0.151$, partial $\eta^2 = 0.041$, $BF_{10} = 0.456$), or interaction effects ($F_{(6.850,287.681)} = 0.814$, $p = 0.574$, partial $\eta^2 = 0.019$, $BF_{10} = 0.013$) were found for the post-learning test.

For an assessment of performance speed, response times significantly decreased across sequential practice blocks, as shown by a main effect of block ($F_{(7.900,995.358)} = 64.246$, $p < 0.001$, partial $\eta^2 = 0.338$, $BF_{10} = 5.881e^{152}$). These performance improvements were similar among age groups, as indicated by the lack of a significant group effect ($F_{(3,126)} = 1.578$, $p = 0.198$, partial $\eta^2 = 0.036$, $BF_{10} = 0.291$) and group x block interaction ($F_{(23.699,995.358)} = 1.337$, $p = 0.130$, partial $\eta^2 = 0.031$, $BF_{10} = 1.21e^{-04}$). During the post-learning test run, there was no significant block effect ($F_{(2.102,264.890)} = 0.404$, $p = 0.678$, partial $\eta^2 = 0.003$, $BF_{10} = 0.012$), indicating a performance plateau was reached. Moreover, consistent with the training run, there were no significant differences among groups (Group main effect: $F_{(3,126)} = 1.099$, $p = 0.352$, partial $\eta^2 = 0.025$, $BF_{10} = 0.318$; Group x Block interaction: $F_{(6.307,264.890)} = 1.148$, $p = 0.335$, partial $\eta^2 = 0.027$, $BF_{10} = 0.032$). Collectively, the assessment of performance across blocks of practice indicates that the four groups acquired the novel motor sequence, as evidenced by the reduction in RTs in the training run. And, we found no credible evidence that the four groups differed in these performance improvements.

### Experiment 1—sequence-specific learning
A learning magnitude measure was calculated for performance speed that reflects the relative difference between the random and sequential task variants at the end of training and therefore affords the assessment of the magnitude of sequence-specific learning (see Fig. 2c). Results revealed a significant group effect ($F_{(3,129)} = 5.440$, $p = 0.002$, $\eta^2 = 0.115$, 95% CI = [0.20, 0.208], $BF_{10} = 20.731$). More specifically, a significantly smaller

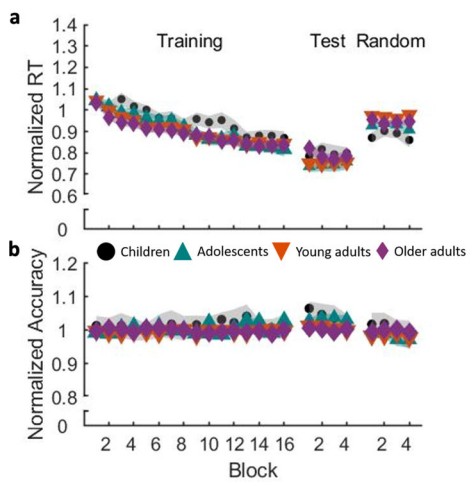
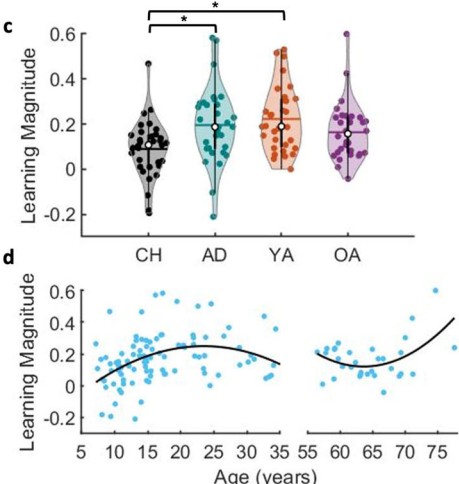

**Fig. 2 | Initial motor sequence learning. a** Average normalized response time (RT) for both task variants. **b** Average normalized accuracy for both task variants. For **a** and **b**, shaded regions represent standard errors of the mean. **c** Average learning magnitude for performance speed per age group. Shaded regions represent the kernel density estimate of the data, colored circles depict individual data, open circles represent group medians, and the horizontal lines depict group means[65]. CH children, AD adolescents, YA young adults, OA older adults. *$p < 0.05$ for pairwise group comparisons. **d** Learning magnitude as a function of age. Quadratic fit from childhood into young adulthood: Learning magnitude = (-0.0008572*age$^2$) + (0.040$^2$1*age) + (−0.2226). Quadratic fit for older adults: Learning magnitude = (0.00157*age$^2$) + (−0.1996*age) + 6.4675. $n = 33, 33, 32$ and 32 for groups of children, adolescents, young adults, and older adults, respectively.

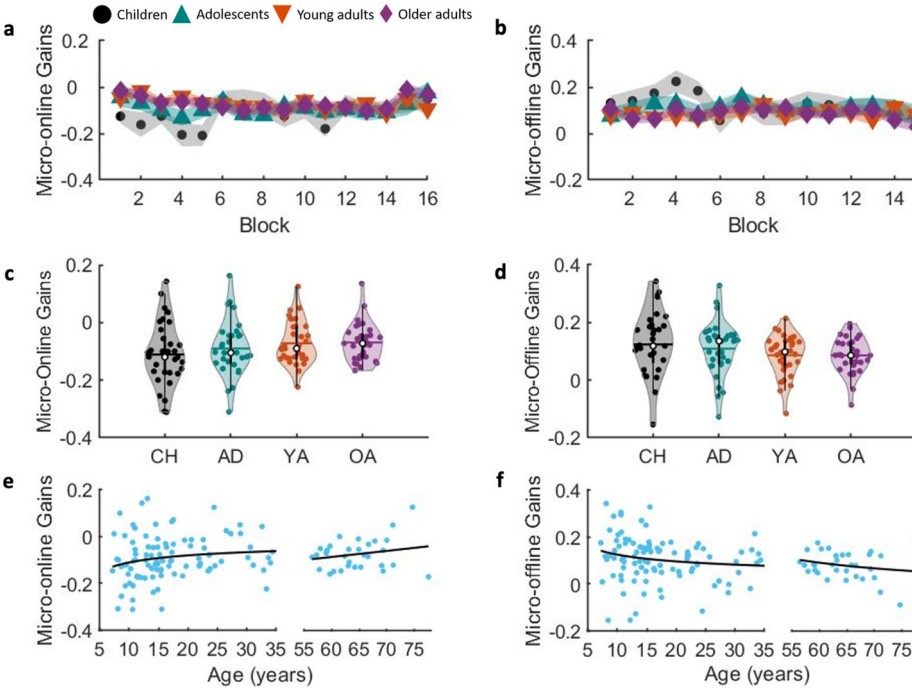

**Fig. 3 | Micro-online (left) and -offline (right) gains. a, b** Depict micro-online and offline gains, respectively, displayed as a function of practice blocks (N children = 31, N adolescents = 33, N young adults = 32, N older adults = 32). **c, d** Contain violin plots of micro-online and -offline, respectively, gains averaged across blocks. Note that pairwise comparisons in these panels included blocks or rest periods in the statistical models. Shaded regions represent the kernel density estimate of the data, colored circles depict individual data, open circles represent group medians, and the horizontal lines depict group means[65]. CH children, AD adolescents, YA young adults, OA older adults. **e, f** show averaged micro-online and -offline gains, respectively, as a function of age from childhood into young adulthood and within older adulthood. Lines of fit childhood into young adulthood: micro-online gains = $(-0.318*age)^{-0.459}$; micro-offline gains = $0.298*exp(-0.377*age)$. Lines of fit within older adulthood: micro-online gains = $(0.003*age) + (-0.238)$; micro-offline gains = $(227.361*age) + (-1.897)$. $n$ = 33, 33, 32 and 32 for groups of children, adolescents, young adults, and older adults, respectively.

sequence-specific learning magnitude was observed in children as compared to adolescents ($p = 0.014$, G = $-0.714$, 95% CI = [$-1.204$, $-0.219$], $BF_{10} = 8.645$) and young adults ($p = 0.001$, G = $-0.970$, 95% CI = [$-1.476$, $-0.457$], $BF_{10} = 124.212$). Follow-up analyses indicated that the difference in learning magnitude could be attributed to significant group differences in RT for the post-learning random run ($F_{(3,126)} = 5.795$, $p < 0.001$, partial $\eta^2 = 0.121$, $BF_{10} = 35.104$). Children demonstrated greater improvements in performance speed from the pre-learning to post-learning *random* runs as compared to young ($p < 0.001$, G = $-0.876$, 95% CI = [$-1.377$, $-0.369$], $BF_{10} = 43.280$) and older adults ($p = 0.022$, G = $-0.591$, 95% CI = [$-1.081$, $-0.098$], $BF_{10} = 3.018$). Accordingly, a portion of the performance improvements across the sequential training blocks in children can be attributed to general learning of the task that is *not* specific to the sequence.

We also conducted exploratory regression analyses with age as a continuous variable to provide a more fine-grained characterization of the age-related trajectories of sequence-specific learning magnitude. Consistent with the differences among *groups*, age was a significant predictor of learning magnitude from childhood into young adulthood (see Fig. 2d; the best fit model was a quadratic function: $R^2 = 0.156$, $F_{(1,95)} = 8.808$, $p = 0.004$). Specifically, learning magnitude increased across childhood, reaching peak values at approximately 24 years of age, and then decreased thereafter. Age was also a significant predictor of learning magnitude within the older adult group, exhibiting an approximate U-shaped trajectory from 55 to 75 years (quadratic model: $R^2 = 0.247$, $F_{(1,29)} = 4.746$, $p = 0.038$).

**Experiment 1—micro-learning across sequential training**
For micro-online learning, the block x group interaction was not significant ($F_{(34.573,1429.030)} = 1.159$, $p = 0.243$, partial $\eta^2 = 0.027$, $BF_{10} = 0.001$; see Fig. 3a). There was, however, a significant block effect ($F_{(11.524,1429.030)} = 2.358$, $p = 0.006$, partial $\eta^2 = 0.019$, $BF_{10} = 0.140$) as well as a non-significant trend for a group effect ($F_{(3,124)} = 2.663$, $p = 0.051$, partial $\eta^2 = 0.061$, $BF_{10} = 0.259$; see Fig. 3c). Although these findings suggest differences in micro-online learning across blocks of practice and among age groups, the corresponding BFs indicate that the data are more consistent with no differences as compared to the alternative (i.e., evidence of differences). Consistent with these results, there was only a non-significant trend for age being a predictor of micro-online gains from childhood into young adulthood (power model: $R^2 = 0.032$, $F_{(1,96)} = 3.141$, $p = 0.080$; see Fig. 3e).

No age-related changes were revealed within the older adult group (power model: $R^2 = 0.045$, $F_{(1,30)} = 1.424$, $p = 0.242$).

The assessment of micro-offline learning showed no rest block ($F_{(11.662,1446.066)} = 1.209$, $p = 0.272$, partial $\eta^2 = 0.010$, $BF_{10} = 2.28e^{-04}$; see Fig. 3b), nor a rest block x group interaction ($F_{(34.985,1446.066)} = 0.836$, $p = 0.739$, partial $\eta^2 = 0.020$, $BF_{10} = 7.098e^{-05}$). There was a non-significant trend for a group effect ($F_{(3,124)} = 2.490$, $p = 0.063$, partial $\eta^2 = 0.057$, $BF_{10} = 0.217$), yet the corresponding BF indicated that the data are more in favor with the null hypothesis as compared to the alternative. Similarly, the characterizations of age-related trajectories revealed that there was only a non-significant trend for age being a predictor of micro-offline gains from childhood into young adulthood (power model: $R^2 = 0.029$, $F_{(1,96)} = 2.873$, $p = 0.093$; see Fig. 3f). No significant age-related changes were revealed within older adulthood (power model: $R^2 = 0.043$, $F_{(1,30)} = 1.362$, $p = 0.252$).

**Experiment 2—participant characteristics, sleep and vigilance**
Group means for participant characteristics, the times of when the experimental sessions were completed, the duration of the offline periods between sessions, self-reported sleep quality and duration for the nights preceding each experimental session, and subjective (SSS) and objective (PVT) measures of vigilance are provided in Table 2. Results from the corresponding statistical analyses and a brief discussion can be found in Supplementary Note 1 and Supplementary Table 2.

**Experiment 2—initial learning dynamics**
Similar to Experiment 1, we assessed the dynamics of learning during the initial training session of Experiment 2. Normalized performance measures are shown in Fig. 4. For brevity, the corresponding statistical results can be found in Supplementary Table 5. In brief, the results from the assessment of the performance improvements during the initial training session of Experiment 2 largely mirrored those presented above for Experiment 1. We found little credible evidence for differences among age groups in the performance changes across training blocks.

**Experiment 2—macro-offline performance gains**
Results on the sequential macro-offline changes in performance (Fig. 5) revealed both offline period ($F_{(1,104)} = 104.357$, $p < 0.001$, partial $\eta^2 = 0.501$, $BF_{10} = 1.037e14$) and group ($F_{(3,104)} = 14.541$, $p < 0.001$, partial $\eta^2 = 0.296$,

**Fig. 4 | Normalized motor performance in Experiment 2.** Average normalized response time (RT; **a**) and accuracy (**b**) for the three experimental sessions and two task variants of Experiment 2. Shaded regions represent standard errors of the mean. Corresponding statistical analyses are in Supplementary Table 5. $n = 27$ in each of the 4 age groups, except $n = 26$ for young adult data in Session 3 and adolescent data in post-test random of Session 3.

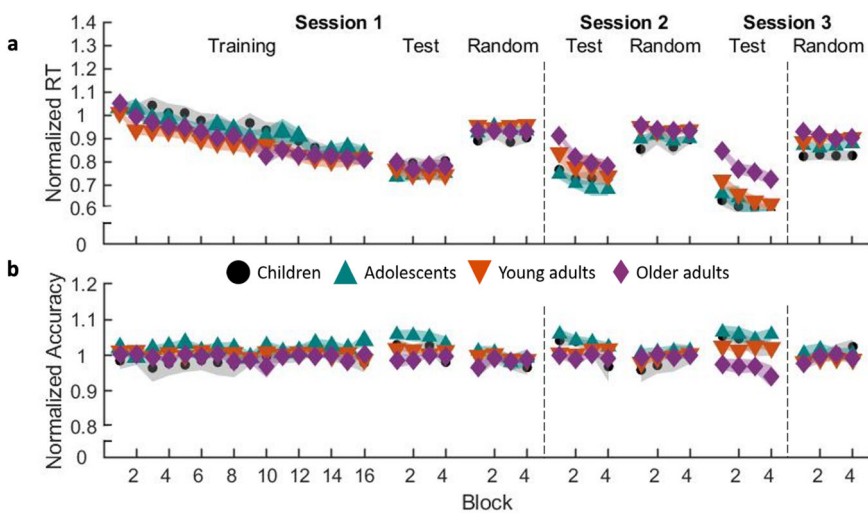

$BF_{10} = 90620.35$) main effects, as well as a significant offline period x group interaction effect ($F_{(3,104)} = 3.382$, $p = 0.021$, partial $\eta^2 = 0.089$, $BF_{10} = 0.384$). Follow-up comparisons revealed significant group effects for both the 5h ($F_{(3,107)} = 5.756$, p = 0.001, $\eta^2 = 0.142$, 95% CI = [0.027, 0.249], $BF_{10} = 28.76$) and the 24h ($F_{(3,107)} = 20.582$, $p < 0.001$, $\eta^2 = 0.373$, 95% CI = [0.216, 0.479], $BF_{10} = 26.800e^{06}$) sequential macro-offline gains. Post-hoc pairwise comparisons indicated that the sequential macro-offline gains after a 5h period of wakefulness were higher in children and adolescents as compared to the young ($p = 0.032$, G = 0.590, 95% CI = [0.049, 1.125], $BF_{10} = 1.943$; $p = 0.053$, G = 0.548, 95% CI = [0.009, 1.082], $BF_{10} = 1.497$, respectively) and older adults ($p = 0.008$, G = 1.382, 95% CI = [0.788, 1.966], $BF_{10} = 3765.434$; $p = 0.015$, G = 1.290, 95% CI = [0.704, 1.866], $BF_{10} = 1289.955$, respectively). It is worth emphasizing that the BFs for the comparisons between the children, adolescents and young adults suggest only anecdotal evidence that the data are in favor of the alternative hypothesis (i.e., evidence for group differences) as compared to the null hypothesis for the 5h offline gains. The children showed higher 24h macro-offline gains in comparison to the young ($p = 0.001$, G = 1.285, 95% CI = [0.699, 1.861], $BF_{10} = 574.756$) and older ($p < 0.001$, G = 2.449, 95% CI = [1.737, 3.147], $BF_{10} = 1.080e^{08}$) adults, and the older adults displayed significantly lower 24h macro-offline gains than all the other age groups (adolescents: $p < 0.001$, G = 1.256, 95% CI = [0.673, 1.829], $BF_{10} = 658.860$; young adults: $p = 0.002$, G = 1.237, 95% CI = [0.656, 1.809], $BF_{10} = 458.448$).

In line with the age group comparisons presented above, our analyses with age as a continuous variable revealed that age significantly predicted the 5 h macro-offline gains from childhood into young adulthood (linear model: $R^2 = 0.068$, $F_{(1,78)} = 5.650$, $p = 0.020$; see Fig. 5b). Age was not a significant predictor of the 5h offline gains within the older adult group (double exponential model: $R^2 = 0.183$, $F_{(1,23)} = 1.72$, $p = 0.203$). Lastly, age significantly predicted the 24h macro-offline gains from childhood to young adulthood (Power model: $R^2 = 0.157$, $F_{(1,78)} = 14.537$, $p < 0.001$; see Fig. 5d) as well as within older adulthood (linear model: $R^2 = 0.136$, $F_{(1,25)} = 3.95$, $p = 0.058$).

### Experiment 2 - link between the offline performance changes on a micro- and macro-timescale

The regression model assessing the relationship between the micro- and 5h macro-offline gains – independent of age group – revealed a significantly positive relationship between these two offline measures ($b = 0.581$, $p < 0.001$, $BF_{10} = 831.113$; see Fig. S9). This suggests that greater performance changes during the short rest periods throughout training were associated with greater performance gains across the 5h offline period of wakefulness following training. We then assessed whether such a relationship differed between pairs of age groups. Young adults exhibited a steeper

slope as compared to children ($b = -1.371$, $p = 0.003$, $BF_{10} = 0.381$), adolescents ($b = -1.287$, $p = 0.002$, $BF_{10} = 0.476$) and older adults ($b = -1.509$, $p = 0.003$, $BF_{10} = 0.475$). However, the reported BFs indicate that the data are more consistent with no differences and these between-group differences were no longer statistically significant if the young adult participant with the extremely negative 5h offline gain (see Fig. 5a) was removed from the analyses.

Micro-offline gains were also significantly and positively related to the 24h macro-offline gains ($b = 0.609$, $p < 0.001$, $BF_{10} = 16736.4$; see Fig. S10). Specifically, greater performance changes during the short rest periods were also associated with greater performance gains across the 24h period following training. This relationship was significantly different between the children and young adults ($b = -0.669$, $p = 0.047$, $BF_{10} = 6.268$). Specifically, whereas young adults exhibited a significant positive relationship between micro- and macro-offline performance gains ($b = 0.647$, $p = 0.007$, $BF_{10} = 6.380$), children did not ($b = -0.023$, $p = 0.90$, $BF_{10} = 0.360$). There is little credible evidence that this relationship between micro-offline and 24h macro-offline gains was different between young adults and the other two age groups (adolescents: $b = 0.126$, $p = 0.679$, $BF_{10} = 0.594$; older adults: $b = -0.217$, $p = 0.556$, $BF_{10} = 0.189$).

## Discussion

The current study examined age-related differences among children, adolescents, young adults and older adults in initial motor sequence learning and motor memory consolidation. Our results revealed that: (1) the dynamics of initial motor sequence learning, as demonstrated by performance improvements during training, were comparable (i.e., not statistically different) across age groups; (2) children exhibited less sequence-specific learning during the initial training session relative to adolescents and young adults; (3) although micro-online and -offline performance changes were the smallest and largest, respectively, in children, there was not sufficient evidence to support the hypothesis of differences among age groups; (4) children exhibited greater changes in performance from the end of initial training to both the 5h and 24h retests, suggesting a developmental advantage in the macro-offline consolidation of a newly acquired movement sequence; and, (5) there was a significant positive relationship between the magnitudes of micro- and macro-offline performance changes.

### Initial performance improvements are comparable across age groups, but children exhibit less sequence-specific learning

On the one hand, our results provided no evidence that performance improvements across training differed among age groups, which was consistent with a subset of the previous literature[5–8,22–26,46]. This finding thus agrees with the lifespan invariance model that has been discussed in

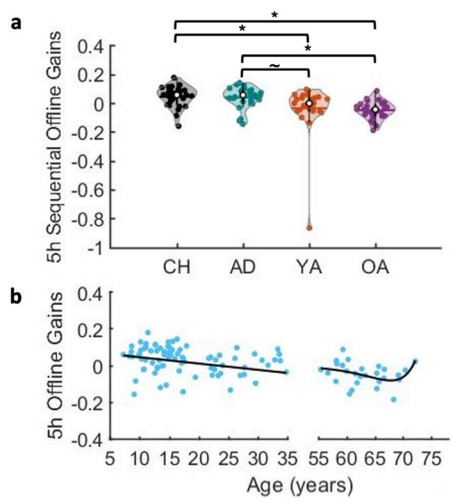
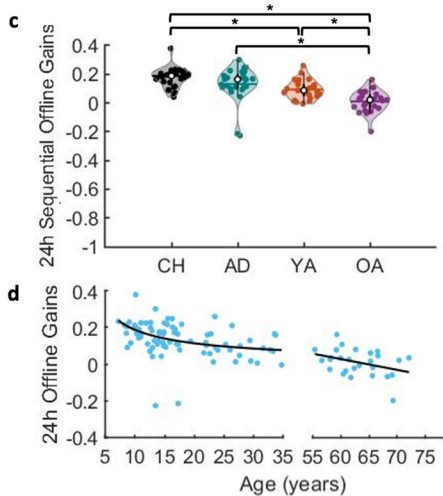

**Fig. 5 | Macro-offline performance gains.** Sequential offline gains across the 5 h (**a**) and 24 h (**c**) offline periods for the four age groups. Shaded regions represent the kernel density estimate of the data, colored circles depict individual data, open circles represent group medians, and the horizontal lines depict group means[65]. CH children, AD adolescents, YA young adults, OA older adults. *$p < 0.05$ and ~$p < 0.10$ for pairwise group comparisons. $n = 27$ in each of the 4 age groups. Five-hour (**b**) and 24h (**d**) sequential offline gains are plotted as a function of age. Lines of fit from childhood into young adulthood: 5h gains = $(-0.003 * age) + 0.075$; 24 h gains = $(0.872*age)^{-0.682}$. Lines of fit within older adulthood: 5h gains = $(1.814e^{-6}$ * exp(0.256 * age)) + $(-1.858e^{-6}$ * exp(0.256 * age)); 24h gains = $(-0.006 * age)$ 0.381. For better visualization of the age-related changes, the scale of the $y$-axes was set to range from $-0.4$ to $0.4$ and thus the young adult with the outlier 5h sequential gain (visible in **a**) is not depicted in (**b**). This individual, however, was included in the fitting procedure. Note that if this individual was excluded from statistical analyses, the pattern of results observed in (**a**) remains largely similar, with the exception that the pairwise difference between young and older adults becomes statistically significant (i.e., older adults exhibit worse offline consolidation over a 5h wake interval as compared to young adults).

previous literature in the context of memory consolidation[29]. On the other hand, our results indicated that children exhibited less sequence-specific learning and thus their performance improvements across sequential training blocks are partially due to general motor improvements (i.e., improvements on the motor task that are *independent* of sequence learning). This finding – which resembles the prototypical developmental trajectory where young adults perform the best and children are progressing towards this optimal performance level - is consistent with a subset of previous literature in which children exhibited smaller sequence-specific motor learning[11,12,47]. The pattern of results may explain some of the heterogeneity that is present in the literature with respect to the development of motor sequence learning. That is, previous research that assessed performance improvements across blocks of practice may report comparable learning between children and young adults (e.g.,[8,20]), whereas studies that afford the extraction of sequence-specific learning may reveal differences between age groups (e.g.,[11,47]). Presumably, the greater general learning that is evident in children is due to this group simply learning how to perform the task itself (i.e., spatially mapping the stimuli on the screen to the appropriate fingers). Future research that aims to isolate and examine sequence learning *per se* could consider incorporating an extended task familiarization phase consisting of additional random runs (i.e., no sequence to be learned). This would allow children to exhaust their general learning of the task itself prior to sequence acquisition. Nevertheless - and as a point of emphasis - even though sequence-specific learning was smaller in children, they still exhibited significant learning of the motor sequence. That is, their sequence-specific learning magnitude was significantly greater than zero (see Supplementary Table 6) and thus a memory trace of the acquired sequence was presumably formed. Interestingly, whereas children showed a smaller learning magnitude, there was no credible evidence that the adolescents differed from young adults. This suggests that sequence-specific learning reaches adult-like levels by the teenage years.

## Micro-offline consolidation during interspersed rest breaks

In line with previous literature[5] and our hypothesis, children exhibited the largest micro-offline performance gains. However, the group main effect within the omnibus group (4 levels) x block ANOVA was a non-significant

trend ($p = 0.063$) and the corresponding Bayes Factors did not provide substantial evidence in favor of age group differences. Specifically, although pairwise comparisons indicated that micro-offline gains were larger in children as compared to both young and older adults ($F_{(1,61)} = 4.829$, $p = 0.032$, partial $\eta^2 = 0.073$ and $F_{(1,61)} = 5.374$, $p = 0.024$, partial $\eta^2 = 0.081$, respectively), the corresponding BFs were 1.109 and 1.411, respectively, and thus indicative of only anecdotal evidence in support of age group differences[41]. It is worth noting here that the assessment of micro-offline gains on non-normalized RT data did reveal significant age group effects, with children exhibiting larger gains than the other 3 age groups (see Supplementary Note 3 and Fig. S6). This analysis, however, was not part of our pre-registration. Nonetheless, the result of our pre-registered analysis of micro-offline gains presented in the main text can be viewed as weaker in comparison to Du et al.[5] One potential explanation could be the specific ages included in the child groups. Specifically, Du et al.[5] found a micro-offline advantage in younger children (i.e., 6-year-olds) as compared to older children (i.e., 10-year-olds) and adults. In the current research, the online nature of our protocol made it difficult to acquire data in such young children (i.e., 6-year-olds). Moreover, as data acquisition in Du et al.[5] was limited to these discrete age groups in (i.e., data from 7–9 year-old children were not acquired), an estimate of when this childhood advantage in micro-offline processing disappears was not possible. We speculate that the differences between children and young adults in our study would have been larger in magnitude if the sample included a greater number of younger children (i.e., 6- to 7-year-olds). Another potential consideration is the considerably longer rest breaks (i.e., 3 min) between practice blocks that Du et al.[5] implemented. In comparison to our 15 second rest intervals, this allowed more time for the recently acquired memory trace to be consolidated over these micro-offline epochs. Of note, recent research in young adults has produced inconclusive results with respect to the impact of the rest period durations. In a probabilistic sequence task, there was no effect of duration (15 s vs. 30 s vs. self-selected) on micro-offline performance changes[48], whereas a temporal gradient (10 s vs. 20 s) was found in resistance to an interfering sequence in a deterministic motor sequence task[49]. In summary, future studies are needed to further examine micro-learning processes in children, with systematic examinations into certain

methodological choices such as the age of the participants, the length of the rest epochs and the choice of sequence task (i.e., probabilistic vs. deterministic).

Although this fast consolidation process occurring over micro-offline intervals has received considerable attention over the last 5 years[43–45,49,50], limited research has examined micro-learning processes in older adults. Our results found no evidence for differences in micro-online and -offline performance gains between young and older adults. Accordingly, whereas older adults have deficits in the macro-offline consolidation processes occurring between two practice sessions[24–26,28], they do not exhibit deficits in the rapid consolidation process that occurs during the rest periods interspersed with blocks of task practice.

Lastly, similar to previous studies that employed the self-initiated, sequential finger tapping task in young adults[5,43–45], our results with the SRTT also provide evidence for a significant contribution of the micro-offline gains to overall learning in all four age groups. That is, performance improvements across blocks of training were largely due to micro-offline – as compared to micro-online – performance gains. However, and independent of the specific task variant, one cannot rule out the effect of fatigue on these results. Gupta and Rickard[51,52] have argued that the magnitude of the micro-offline gains in young adults can be attributed to the build-up of fatigue or reactive inhibition as a function of task practice within blocks. Specifically, it was suggested that performance deteriorates across practice within a block due to reactive inhibition and thus micro-offline performance gains are inflated due to this performance variable that can be considered independent of learning and memory processes. However, it is worth noting that this view cannot account for recent neuroimaging data showing reactivation of sequence learning-related patterns of hippocampal activity during the interspersed rest periods[43,44,50]. These neuroimaging data then suggest that micro-offline epochs afford an opportunity for a rapid memory consolidation process. Additional research is necessary to definitively conclude whether these micro-offline performance gains are reflective of a memory consolidation process or are simply attributed to reactive inhibition.

## Children exhibited a developmental advantage in macro-offline consolidation

Our results revealed larger sequential performance gains across both the 5h and 24h macro-offline periods in children. While the young adults showed a performance maintenance (i.e., offline gains that were not significantly different from zero; see Supplementary Table 7) across the 5h interval, children and adolescents significantly improved their performance on the sequential task. Interestingly, children also demonstrated higher offline gains across the 24h period as compared to adults. While the enhanced 5h consolidation in children is in line with the findings of Ashtamker & Karni[14] that showed an accelerated consolidation in children across a one-hour offline period, the greater 24h gains stand in contrast to previous research that demonstrated impaired overnight consolidation in children [refs. 13,15; see below for expanded discussion on sleep-related consolidation].

It could be argued that the larger macro-offline processing in children could be explained by performance levels reached at the end of the training session[53,54]. For example, perhaps children showed greater macro-offline gains due to a continuation of initial learning rather than the offline processing of previously acquired information. This potential explanation, however, is not in line with our data. Notably, during the post-training test phase, there was an absence of a block effect in all groups, including in the children. This indicates that all groups reached a performance plateau at the end of the first session and thus macro-offline gains do not appear to be the result of continued learning. Along the same lines, one could speculate that the young adults reached a performance ceiling effect, and thus their potential to further increase their performance over macro-offline intervals was saturated (see Fig. S4 for a plot of absolute reaction times). Again, this is not consistent with our data. Notably, there was a significant session effect in sequential offline gains, whereby gains were larger in the 24h retest as compared to the 5h retest. Within-group comparisons revealed that this

session effect was present for all age groups, including young adults. This suggests that a performance ceiling effect was not reached and thus this explanation cannot fully account for our results. Last, one could raise the question whether perhaps the macro-offline performance gains are inflated in children due to non-sequence-specific improvements (i.e., improvements due to motor task familiarization as opposed to sequence learning). However, the children did not show greater macro-offline gains on the random task variant (see Supplementary Note 5 and Fig. S11). Moreover, and to assess this possibility further, we examined the offline changes in the sequence-specific learning magnitude (i.e., difference between the learning magnitude (i.e., sequence vs. random) of the 5h and 24h retest sessions and the learning magnitude of the first session; see Supplementary Note 5 and Fig. S11). These results revealed significant main effects of offline period and group, but no offline period x group interaction effect. Post-hoc pairwise comparisons indicated that children exhibit enhanced sequence-specific offline gains—across both offline periods—in comparison to young and older adults. This finding is consistent with the assessment of the sequential macro-offline gains presented in the main text (Fig. 5). Altogether, these data suggest that the offline changes in children likely involved the strengthening of their sequential memory and that this strengthening was greater as compared to young and older adults.

Offline performance gains on the 24h retest were largest in children, a result that was unexpected based on previous literature[13,15]. Specifically, children have typically exhibited impaired consolidation over post-learning intervals that include sleep as observed in young adults. It is worth noting that most of the previous studies comparing sleep and wake conditions in children employed an AM/PM design. In this design, participants acquire the motor sequence at a different time of day based on the condition and then are retested after a delay of 12 h. For example, the wake (i.e., AM-PM) group initially learns the sequence in the morning and then is retested following 12 h of wakefulness. The sleep (i.e., PM-AM) group completes initial learning in the evening and this is retested after a 12 h interval that included a night of sleep. Although informative, these designs are sensitive to time-of-day effects. And there is evidence that the time of initial practice can affect performance during acquisition and offline consolidation in young adults[55,56]. Specifically, Kvint et al.[56] found a greater sequential knowledge acquisition but similar 12h consolidation in an AM-trained as compared to a PM-trained group. Furthermore, Truong et al.[55] showed an enhanced consolidation when the training and retest were completed at 3 pm (as compared to 10 am). These findings indicate time-of-day effects for the acquisition and consolidation outcomes in young adults. Although previous research that employed AM/PM designs in children did not find differences in motor skill acquisition between morning and evening groups[8], the impact on offline consolidation processes has yet to be systematically and exhaustively examined. Although certainly speculative, it is possible that the impaired overnight consolidation that previous studies have found in children could be – at least partially – attributed to the choice of experimental design. An alternative, albeit related explanation, is that the beneficial effect of sleep requires additional time to emerge at the behavioral level and thus that these previous studies did not show overnight gains due to their retest immediately after the sleep episode (i.e., 12h after initial learning). Consistent with this explanation, Desrochers et al.[19] found that a post-learning nap facilitated the consolidation of a recently acquired motor sequence in pre-school children, but only after an extended period of time that included a night of sleep. Additional research is necessary to further disentangle these potential explanations.

## Potential neural correlates of enhanced macro-offline processing in children

An open question is: What are the neural correlates of the greater macro-offline performance gains observed in children? Based on (a) recent evidence in young adults demonstrating that the patterns of activity observed in the hippocampus and putamen during motor learning are spontaneously reactivated in macro-offline intervals[57]; and, (b) the known role of reactivation processes in memory consolidation and longer-term retention[58,59], it

is tempting to speculate that the childhood advantages observed in this research can be attributed to enhanced reactivation of learning-related patterns of brain activity. This possibility is pure speculation at this point but deserves attention in future research.

## Older adults exhibit impaired overnight consolidation

On the other end of the lifespan, our results showed similar 5h but smaller 24h offline gains in the older as compared to the young adults. The comparable 5h offline gains are in line with previous studies that showed a similar consolidation between young and older adults across an offline period of wakefulness[24,25]. Similarly, the smaller 24h offline gains observed in the older adults are in line with several studies that demonstrated impaired overnight consolidation with aging[25,26,28]. These studies have largely attributed this deficit to age-related decreases in sleep quality and quantity, as well as alterations in the sleep-related electrophysiological markers of plasticity such as sleep spindles and slow waves[22,24] (but see[3,23,25] for discussion of alternative explanations). Whereas our older adult participants did report shorter sleep duration as compared to both children and adolescents, no credible evidence for differences in sleep quality were found. However, it is important to note that these measures were self-reported and previous research has shown that older adults tend to overestimate their sleep quality[60]. Nonetheless, our results replicate the often demonstrated finding that aging negatively impacts overnight consolidation of recently acquired motor sequences.

## Limitations

There are some limitations and methodological considerations of the current research that warrant further attention. *First*, these experiments were conducted online and thus were not monitored by researchers in real-time. Although we cannot completely discard the possibility that the online nature influenced our results, it is worth nothing that previous studies that compared online and in-person experiments in children[61] and young adults[49,62] found comparable results between these study modalities. For instance, Cubillos et al.[62] showed similar learning curves for the performance of the finger-tapping task between a supervised-lab and unsupervised-online young adult group. Additionally, Bönstrup et al.[49] obtained similar micro-offline learning in young adults through an online crowdsourcing platform. It is also worth noting that our protocol was not implemented in common crowdsourcing platforms (e.g., Mechanical Turk) that typically acquire full datasets within hours of posting an experiment and have come under scrutiny for unreliable data[63]. Furthermore, although there are drawbacks to employing web-based data acquisition, a significant advantage is the recruitment from a large and broad pool of participants, enabling us to include individuals that otherwise may not have been able to complete our traditional in-person studies.

Second, and specific to experiment 2, each session ended with a pseudorandom run. The implementation of this random run is often used in sequence learning studies and has the benefit of assessing sequence specificity of performance improvements[5,6,11,47,64]. However, we cannot rule out the possibility that the random runs at the end of each session interfered with the macro-offline processing of the sequential motor memory.

Third, explicit awareness was *not* assessed at the end of any session or experiment. No explicit recognition or recall test was implemented due to the online nature of the experiment and the potential interference with the offline processes in Experiment 2. Nevertheless, the lack of such a test impedes our ability to draw any conclusions regarding age groups differences in explicit awareness and how such a difference may affect the learning and memory results.

Fourth, it is important to note that, due to concerns surrounding the acquisition of quality data from young children, our minimum age for this research was 7 years. Moreover, this youngest age was relatively undersampled in our experiments ($n = 3$ and 1 7-year-olds included in analyses of Experiments 1 and 2, respectively). Due to difficulties with completing the online motor task, several 7-year-old children had to be excluded from data analyses. Future studies could consider implementing certain measures (e.g.,

monitoring through videocall) to increase the number of young children that successfully complete the experiments.

Fifth, and as outlined in our pre-registration, the age groups were operationally defined based on age (i.e., children and adolescents were between 7 and 12, and 13 and 17 years, respectively), which does not consider the participant's pubertal status. Our questionnaire battery did ask the parents to indicate whether their child exhibited signs of puberty onset at the time of participation. Note that all participants 13 years and older (i.e., classified as an adolescent based on age) were reported to having started puberty. And only a minimal number of child subjects (6 in each of the experiments) were considered to have initiated puberty. Given the relatively few subjects that would be re-classified based on parental report of puberty onset, we do not anticipate our results to substantially change if pubertal status – as opposed to age - was used to demarcate childhood and adolescence.

## Conclusions

The current study examined initial motor sequence learning and the time course of motor memory consolidation across the human lifespan. Our results demonstrated comparable dynamics of performance improvements during the initial learning session, yet 7- to 12-year-old children exhibited smaller sequence-specific learning. Children demonstrated the largest performance gains over macro-offline intervals, suggesting that children exhibit a developmental advantage in the offline processing of recently practiced movement sequences. The neural underpinnings of this childhood advantage are not yet known and thus future research is warranted to characterize them.

## Data availability

The source data used for the results presented in this text as well as the raw data are publicly available without restriction on Zenodo (https://doi.org/10.5281/zenodo.8274118).

## Code availability

Customized data processing codes in MATLAB are publicly available without restriction on Zenodo (https://doi.org/10.5281/zenodo.8274118).

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

## Acknowledgements

This research was supported by internal funds at the University of Utah. The funders had no role in study design, data collection and analysis, decision to publish or preparation of the manuscript.

## Author contributions

A.V.R.: Methodology, Software, Formal Analysis, Investigation, Resources, Data curation, Writing—original draft, Visualization, Project administration. G.A.: Methodology, Writing—reviewing and editing, Supervision. R.D.B.: Methodology, Writing—reviewing and editing. B.R.K.: Conceptualization, Methodology, Resources, Writing—reviewing and editing, Supervision, Project administration.

## Competing interests
The authors declare no competing interests.
