## [Peer Review File · Communications Psychology]

2nd Nov 23

Dear Dr. King,

Thank you for your patience during the peer-review process. Your manuscript titled "Children exhibit a developmental advantage in the offline processing of a learned motor sequence" has now been seen by 3 reviewers, and I include their comments at the end of this message. They find your work of interest, but raised some important points. We are interested in the possibility of publishing your study in *Communications Psychology*, but would like to consider your responses to these concerns and assess a revised manuscript before we make a final decision on publication.

We therefore invite you to revise and resubmit your manuscript, along with a point-by-point response to the reviewers. Please highlight all changes in the manuscript text file.

Editorially, we consider a few issues raised by reviewers to be critical, for example the request for additional analyses including sensitivity analyses, the differences between SRT and finger sequencing task and their relevance with motor learning and consolidation, task details and individual differences in consolidation effect, and to use Bayesian stats or equivalence tests to support the claims from the non-significant findings.

For manuscripts that report null results, we require the following:

- Evidence that the study is sufficiently powered to detect the smallest theoretically or pragmatically meaningful effect
- Bayes Factors or equivalence tests to interpret the null results
- Appropriate language to describe the results. (There is no statistical test that can demonstrate absence of an effect. Statements such as 'There is no difference between x and y.' or 'X does not affect Y.' must be revised to read 'We found [no/little] credible evidence of a difference between x and y.' or 'We found [no/little] credible evidence that X affects Y.')

Please also clarify if any data collection occurred prior to your preregistration.

Please note that your revised manuscript must comply with our formatting and reporting requirements, which are summarized on the following checklist: Communications Psychology formatting checklist and also in our style and formatting guide Communications Psychology formatting guide .

Please use the following link to submit your revised manuscript, point-by-point response to the referees' comments (which should be in a separate document to any cover letter) and the completed checklist:

[link redacted]

** This url links to your confidential home page and associated information about manuscripts you may have submitted or be reviewing for us. If you wish to forward this email to co-authors, please

delete the link to your homepage first **

Please do not hesitate to contact me if you have any questions or would like to discuss these revisions further. We look forward to seeing the revised manuscript and thank you for the opportunity to review your work.

Best regards,

Xiaoqing Hu

Xiaoqing Hu, PhD
Editorial Board Member
Communications Psychology
orcid.org/0000-0001-8112-9700

EDITORIAL POLICIES AND FORMATTING

Editorial Policy: Policy requirements (Download the link to your computer as a PDF.)

* **CODE AVAILABILITY:** All Communications Psychology manuscripts must include a section titled "Code Availability" at the end of the methods section. In the event of publication, we require that the custom analysis code supporting your conclusions is made available in a publicly accessible repository; at publication, we ask you to choose a repository that provides a DOI for the code; the link to the repository and the DOI will need to be included in the Code Availability statement. Publication as Supplementary Information will not suffice. We ask you to prepare code at this stage,

to avoid delays later on in the process.

* DATA AVAILABILITY:

All Communications Psychology manuscripts must include a section titled "Data Availability" at the end of the Methods section or main text (if no Methods). More information on this policy, is available at <http://www.nature.com/authors/policies/data/data-availability-statements-data-citations.pdf>.

At a minimum the Data availability statement must explain how the data can be obtained and whether there are any restrictions on data sharing. Communications Psychology strongly endorses open sharing of data. If you do make your data openly available, please include in the statement:

We recommend submitting the data to discipline-specific, community-recognized repositories, where possible and a list of recommended repositories is provided at <http://www.nature.com/sdata/policies/repositories>.

If a community resource is unavailable, data can be submitted to generalist repositories such as figshare or Dryad Digital Repository. Please provide a unique identifier for the data (for example a DOI or a permanent URL) in the data availability statement, if possible. If the repository does not provide identifiers, we encourage authors to supply the search terms that will return the data. For data that have been obtained from publicly available sources, please provide a URL and the specific data product name in the data availability statement. Data with a DOI should be further cited in the methods reference section.

REVIEWERS' EXPERTISE:

Reviewer #1 Motor memory

Reviewer #2 Motor memory, consolidation

Reviewer #3 Motor memory, consolidation, development

REVIEWERS' COMMENTS:

Reviewer #1 (Remarks to the Author):

Van Roy and colleagues examined the developmental trajectories of motor sequence learning across a broad age range spanning from 7 and 75 years old. They found that children exhibited an advantage in the offline processing, at both the micro- and macro-time scale, of recently acquired motor sequences. This study was well-motivated, experiments were pre-registered, the analyses were sound, and the paper was well-written and easy to follow. I have several comments that I hope the authors can address.

I find myself frustrated and perplexed by the specific aspects of motor sequence learning studied in this paper. The authors defined motor sequence learning as “Participants acquire a novel series of interrelated actions (line 38)”, a definition with which I agree. Motor sequence learning as such has often been studied through the serial reaction time (SRT) task where participants respond to visual cues that follow a pattern and the existence of the pattern is unknown to them. However, a significant portion of the citations about micro-online and micro-offline learning in this paper focused on the finger sequencing task. In this task, participants know the sequence before they perform the task. What they practice and improve is the execution of known motor sequences in terms of speed and accuracy. This is distinct from learning a sequential order of interrelated actions in the SRT task as employed in this paper. In particular, when referring to micro-offline learning, the authors cited Bonstrup et al, 2019 and others who used the finger sequencing task. However, other studies (e.g., Du et al 2017) used the SRT task. The conflation of these two tasks becomes evident when the authors introduce the background on the differences in sequence learning between children and adults, as well as in the literature on offline learning in older populations.

This issue is not specific to the authors but rather reflects a broader trend in the field, where these two distinct tasks are often conflated. I believe that clarifying the introduction of this background information could greatly benefit not only the readers but also the field of motor sequence learning (i.e., in particular when studying the neural basis of motor sequence learning) as a whole. It is challenging, but I hope the authors can make the background introduction clearer.

The finger sequencing task is more generally explicit compared to the SRT task. In particular, the finger sequencing task shows a strong manifestation of sleep-dependent offline gains, while the macro-offline gains in the SRT task are more time-dependent rather than sleep-dependent. The task used in this current paper falls somewhere in between the finger sequencing task and the SRT task. Participants knew that a sequence pattern existed but did not know what the pattern was. It is important to establish a clear rationale for choosing this particular task. How do the findings reported in this study, either consistent or inconsistent with previous research, relate to the distinctions between these experimental tasks?

It is unclear whether each individual was assigned a unique sequence to learn, or whether the same sequence was used for all participants. The same question applies to the random sequence. Were there common segments (e.g., triplets) shared by these two sequences? The magnitude of sequence learning was measured by the difference in RT between the learned sequence and the random sequence. It is possible that adult participants acquired the entire sequence as one (especially when they were told there was a pattern before they started the task, so presumably they attempted to and could easily figure out the specific eight-element sequence at the end of learning), while children may learn just some short segments. This could render response speed faster for the common segments between the learned sequence and the random sequences in children, which possibly explains why children responded faster in the random sequences (figure 2A) and thus lower specific sequence learning magnitude. Clarification of these details is crucial for a comprehensive understanding of the study's results.

A related comment is whether the specific sequence learning difference between children and adults is attributed to the explicit awareness of the existence of the sequence pattern. It has been reported that trying to figure out the pattern of a complex sequence could hinder sequence learning (Fletcher, Zafiris, Frith, Honey, Corlett, Zilles, Fink, 2004, Cerebral Cortex). Although the sequence used by the authors was relatively simple to adults, it could still pose a significant challenge for

children if they tried to puzzle out the sequence order.

Each sequence consisted of only eight elements. This means that adult participants may become aware that once they respond with a certain finger, they would not need to press that finger for a few subsequent trials. This is different from traditional SRT tasks in which each stimulus appears twice in a single repetition of a sequence. Do you think this might be the reason why adults demonstrated greater online learning especially that participants were told the existence of the sequence pattern? It would be helpful if the authors measured the explicit knowledge of the acquired sequence after participants finished the first session, but I think this might not be feasible because participants needed to perform the 5-hr and 24-hr retests.

It is a bit hard to follow why RT needed to be normalized when it was used to quantify specific sequence learning. The difference in RT between the learned sequence and the random sequence at the end of session 1 already accounted for the general motor improvement.

The authors discussed the fatigue effect as the alternative possible explanation of the micro-offline gain. However, I did not quite follow why the finding that micro-offline gain is related to hippocampal activities rejects the fatigue explanation. These two cited studies used the finger sequencing task in adult populations. It is still very possible that fatigue accumulates when children keep performing the same SRT task. In addition, the online gains for all groups were negative (Figure 3C), indicating that RT became slower within each block, could this be explained by the fatigue effect? The authors should refrain from downplaying this fatigue explanation.

In the statistical analyses section, it says that repeated measures ANOVA was used, while in the results section, mixed effect models were used.

Were the post hoc analyses corrected for multiple pairwise comparisons? It was not specified in the methods section.

The random blocks at the end of day 1 were not described in the Figure 1 legend.

Reviewer #2 (Remarks to the Author):

This is a study of motor sequence learning using the SRTT. The authors trained 4 groups of participants: children, adolescents, adults, and older adults. They were trained before and after a sleep interval, and also a shorter interval, and consolidation was examined. The main findings are a) consolidation happens both across the 'microinterval' and b) the children showed more consolidation across both intervals, as well as across the micro rest epochs during training.

The study addresses a valid question, since the literature has suggested that children do not show sleep dependent consolidation on this type of task. The authors suggest several possible explanations for the fact that their results are out of keeping with this - including a potential criticism of the design in past studies, which compared sleep to wake.

The study is well written and very clear, with a careful description of methods and results.

Overall, it seems clear that the study deserves publication. I do not find it especially exciting, yet I cannot find any errors or deficits that clearly need amendment either.

Reviewer #3 (Remarks to the Author):

The empirical study and the manuscript contain fascinating data. The issue of memory consolidation is rarely examined from a developmental perspective. Beyond the developmental curve, an important and exciting result is that ultra-fast consolidation can predict longer consolidation processes.

Comments and suggestions:

1) The authors employed a visuomotor task in their studies. In these types of tasks, participants can acquire proficiency even when solely observing the screen without physically pressing any buttons. In light of the brief review by Pedraza et al. (2023) "Nomen est omen: Serial reaction time task is not a motor but a visuomotor learning task," *European Journal of Neuroscience*, Volume 58, Issue 4, on pages 3111-3115, it is suggested that the term "visuomotor learning" be incorporated into the title and manuscript.

2) Prior to the authors' study, only one investigation had explored memory consolidation from a lifespan perspective: Tóth-Fáber et al. "Lifespan developmental invariance in memory consolidation: evidence from procedural memory," *PNAS Nexus*, Volume 2, Issue 3, in 2023, pgad037. While the authors correctly cited this study, it is worth noting that in the discussion section, they erroneously refer to it as concerning initial learning, when in fact, it is a study on memory consolidation. Given that this study constitutes the sole prior research related to the authors' work, the introduction should be more closely aligned with the findings of this study. This adjustment does not diminish the significance of the authors' research, as it pertains to a different sequence type and contains numerous other compelling results. However, it can enhance the manuscript's overall coherence and relevance.

3) One of the most thrilling findings is the ability of ultra-fast consolidation to serve as a predictor for longer-term consolidation. Within this context, it is pertinent to address three critical issues that warrant dedicated paragraphs in the discussion. Firstly, we should delve into the question of individual differences. An essential inquiry pertains to why a substantial number of experimental participants exhibit mini offline consolidation while others do not. The second issue that demands careful consideration is the choice between deterministic and probabilistic sequences. It is worth exploring how this selection may impact the results and their implications. Lastly, the third issue revolves around the duration of the mini-consolidation phase. It is crucial to investigate whether the length of this mini-consolidation phase holds any significance and, if so, to what extent. These discussions are intertwined with two relevant articles: Quentin et al (2021) "Statistical learning occurs during practice while high-order rule learning during rest period." *NPJ Science of learning and Szucs-Bencze et al. (2023). Manipulating the rapid consolidation periods in a learning task affects general skills more than statistical learning and changes the dynamics of learning. *Eneuro*, 10(2).*

4) Related to the previous: Another thought that crossed my mind is that it would be intriguing to examine the relationship between the online and offline consolidation indicators, as well as the micro offline and macro offline regressions. It would be fascinating to explore these group differences (children performing better offline and worse online) at the level of interindividual differences (those who excel in offline performance but struggle online).

5) The participant description section is not entirely clear to me. If I understand correctly, the authors were able to involve 224 people, but according to the sample size calculator, only 130

people's data were used in Experiment 1, and in Experiment 2, there were only 108 people in the end. Is there an overlap between the two samples? Were all 224 people measured in total? Did everyone complete the learning session here? And did some of the participants only take part in Experiment 2? So, authors could describe this a bit more clearly.

6) I think it's problematic that the learning phase ends with a random block. These random blocks can totally change the results compared to a study without random blocks. The authors mentioned it as a limitation. Since it is a crucial point, I would elaborate more on this issue in the manuscript.

7) The results of Experiment 2 say that the initial learning session results are similar to those in Experiment 1, meaning there's no difference in learning. However there was a difference in learning in Experiment 1 (even though it may be more related to genskill than sequence-specific learning).

8) In children, sequence-specific learning in the learning phase was influenced by motor learning (even though the performance was normalised to baseline RT), could this not play a role in macro consolidation? Although there is analysis for random macro-offline gain in the supplementary, there are in principle the same age groups.

9) The study has three 7-year-olds in Experiment 1 and one in Experiment 2. The authors mentioned it in the limitations. But for me, it's a bit questionable whether it would have been better to exclude them, as such a small number of participants doesn't really represent 7-year-olds well.

10) The Sequence-specific learning was double-normalized because first, all blocks were plotted against the baseline data, and then when the learning index was calculated (random minus sequence)/random, the data were again adjusted to the random. I find it particularly strange that the results for the learning indicator are not repeated in the case of non-standardized RTs.

Additionally, their normalization procedure is intriguing to me, and I haven't come across it being used before. But this may be good. Question: how much does this affect the results? Maybe this learning indicator should not be divided by the baseline? It would be nice to look at it anyway and compare it to the double normalized one.

11) Related to the previous: the authors say that the raw data results are included in the supplementary, but as far as I can see, they only report average RT/ACC results per period, so separate for sequence and separate for random blocks - not interesting in my opinion and unnecessary to justify that children are slower at each stage. What would be more interesting/useful would be how the learning indicators change on the raw data compared to the standardized data. So, general skill/seq learning and micro online/offline consolidation should be shown on rawdata (if for no other reason than to be comparable with other research). In Appendix 5, the analyses in the supplementary provide some answers to the sequence-specific offline vs general skill offline problem, but not completely.

12) Exp. 1, the micro-online and micro-offline indicators are not specific to seq-specific learning because they actually only compare block start vs. block end RT (micro-online) and block end vs. block start RT (micro-offline), which, of course, includes sequence, but also includes general skill learning.

13) In Exp. 2, the "sequential offline changes" were only calculated on the sequence blocks, not combined/corrected with random blocks, so the same problem arises: it includes both sequence learning and general acceleration...

14) It would also be worth doing more precise analyses to be able to say more about sequence-specific learning, e.g., at the very least, authors should put the indicators calculated for seq and random blocks into the same ANOVA to see if there is an interaction between the two... Authors should discuss the Quentin's article (see above), because it dissects general skill and seq-specific, even if only for adults...

15) Authors interpret non-significant results as "similar performance", which in principle should not be allowed. Authors should carry out Bayesian analysis if they want to interpret non-significant

results. Bayesian analysis warranted anyway.

Minor:

Line 23: n=238 datasets – s?

Line 182 Table 2 - I suggest to include the p values (from supplementary) to see the statistical comparison between the age groups.

Line 567-570: It's not very clear what is presented here. This should be a Group x micro-Offline interaction, right? Where's the interaction p value?

ID number of the original manuscript: COMMSPSYCHOL-23-0284**Manuscript title: “Children exhibit a developmental advantage in the offline processing of a learned motor sequence”**

We would like to thank the reviewers for their thorough reading and constructive feedback. We have addressed their comments in the detailed responses below. All changes listed in this letter are highlighted in yellow in the revised version of the main text and supplemental material. We believe that the manuscript has substantially improved and hope that the reviewers find our changes satisfactory.

We wanted to first highlight and explain changes that were made to the manuscript that were not in response to specific comments from the reviewers or the editor. These changes are also marked by yellow highlights in the main text and supplemental files.

- In accordance with the author guidelines for *Communications Psychology*, we added Data and Code Availability statements on lines 881 – 886. These statements also include the DOIs to the publicly available dataset and MATLAB code. We also added Author Contributions on lines 887-892.
- During the revision process, we realized the clarity of the participant exclusion criteria could be improved. Accordingly, we altered the text in this section (please see lines 135-158).
- We also recognized that the figure captions for our violin plots did not provide sufficient details. Accordingly, the following descriptive information was added to the captions of all figures that contain violin plots (e.g., Figures 2, 3 and 5 in the main text): “Shaded regions represent the kernel density estimate of the data, colored circles depict individual data, open circles represent group medians, and the horizontal lines depict group means.” These captions also now cite the toolkit used to generate the plots (Bechtold et al., 2021).
- Regrettably, we noticed that the data from the pre-learning random run of a single older adult participant in Experiment 2 was incorrect. We corrected this error and then re-ran all analyses that were affected by this error and generated new figures. The output from the updated statistical analyses is on lines 561 – 627. Importantly, the main results and conclusions remained unchanged.

REVIEWER #1:

Comment 1: Van Roy and colleagues examined the developmental trajectories of motor sequence learning across a broad age range spanning from 7 and 75 years old. They found that children exhibited an advantage in the offline processing, at both the micro- and macro-time scale, of recently acquired motor sequences. This study was well-motivated, experiments were pre-registered, the analyses were sound, and the paper was well-written and easy to follow. I have several comments that I hope the authors can address.

Author response: We thank the reviewer for the positive assessment of our article.

Comment 2: I find myself frustrated and perplexed by the specific aspects of motor sequence learning studied in this paper. The authors defined motor sequence learning as “Participants acquire a novel series of interrelated actions (line 38)”, a definition with which I agree. Motor sequence learning as such has often been studied through the serial reaction time (SRT) task where participants respond to visual cues that follow a pattern and the existence of the pattern is unknown to them. However, a significant portion of the citations about micro-online and micro-offline learning in this paper focused on the finger sequencing task. In this task, participants know the sequence before they perform the task. What they practice and improve is the execution of known motor sequences in terms of speed and accuracy. This is distinct from learning a sequential order of interrelated actions in the SRT task as employed in this paper. In particular, when referring to micro-offline learning, the authors cited Bonstrup et al, 2019 and others who used the finger sequencing task. However, other studies (e.g., Du et al 2017) used the SRT task. The conflation of these two tasks becomes evident when the authors introduce the background on the differences in sequence learning between children and adults, as well as in the literature on offline learning in older populations.

This issue is not specific to the authors but rather reflects a broader trend in the field, where these two distinct tasks are often conflated. I believe that clarifying the introduction of this background information could greatly benefit not only the readers but also the field of motor sequence learning (i.e., in particular when studying the neural basis of motor sequence learning) as a whole. It is challenging, but I hope the authors can make the background introduction clearer.

The finger sequencing task is more generally explicit compared to the SRT task. In particular, the finger sequencing task shows a strong manifestation of sleep-dependent offline gains, while the macro-offline gains in the SRT task are more time-dependent rather than sleep-dependent. The task used in this current paper falls somewhere in between the finger sequencing task and the SRT task. Participants knew that a

sequence pattern existed but did not know what the pattern was. It is important to establish a clear rationale for choosing this particular task. How do the findings reported in this study, either consistent or inconsistent with previous research, relate to the distinctions between these experimental tasks?

Author response: We thank the reviewer for these interesting comments about motor sequence task variants. Our response is structured based on the following 5 points that we extracted from the reviewer's comment.

a) Definition of motor sequence learning and multiple task variants

We agree that multiple task variants are routinely employed in the literature (e.g., see Inset 1 of our review paper that highlights these variants; King et al., 2017). As noted by the reviewer, there are several key distinctions among the common motor sequence task variants. First, each response in the serial reaction time task (SRTT) is cued by a visual stimulus, whereas movements in the finger sequencing task are self-initiated. Second, the sequence to perform is explicitly known or shown in the finger sequencing task, whereas the instructions in the SRTT may be explicit or implicit. Importantly, in both task variants, participants still learn to complete a series of previously unrelated finger movements in a specific order. Accordingly, the definition of motor sequence learning adopted in our paper (i.e., processes by which individuals acquire a novel sequence of interrelated actions) applies to both the SRT (employed in the current study) and finger sequencing tasks. We do, however, agree with the reviewer that readers may benefit if the introduction introduced these different task variants. It would also allow us to clarify when specific results in the literature appear to be specific to a single variant (see part (b) below). Accordingly, our revised introduction now describes the two task variants that are commonly used to assess motor sequence learning (lines 37-43 and copied below).

“MSL has been extensively studied with multiple task variants, including the serial reaction time task (SRTT; i.e., participants respond to visual cues presented in a repeating order) and the finger tapping task (i.e., participants repeatedly reproduce a known/shown sequence of finger movements in a self-initiated manner). There has been extensive previous research that has employed these MSL task variants in healthy young adults (~18-35 years), facilitating the development of a framework that characterizes the time course of sequence learning in this age group (see Albouy et al., 2013; Doyon et al., 2018; King et al., 2017; Krakauer et al., 2019 for reviews).”

b) Micro-offline processes in different task variants

The reviewer specifically referenced the literature on micro-offline learning processes, noting that the work in young adults (Bönstrup et al., 2019, 2020; Buch et al., 2021; Gann et al., 2023; Jacobacci et al., 2020) has employed the FTT whereas the only paper to date in children (Du et al., 2017) employed the SRTT. Our revised version of the introduction now explicitly states the task used when referencing the work in children (lines 63-66 and copied below).

“Specifically, Du et al. (2017) employed a stepping version of a SRTT and found that young children (i.e., 6-year-olds) showed higher micro-offline gains as compared to older children (i.e., 10-year-olds) and young adults.”

To ensure readers are aware that previous research examining micro-offline learning in young adults has been limited to the finger tapping task, we have also modified the corresponding section of the discussion (lines 712-717 and copied below). This updated text also serves to contextualize these results with respect to task variants, as suggested by the reviewer.

“Lastly, similar to previous studies that employed the self-initiated, sequential finger tapping task in young adults (Bönstrup et al., 2019; Buch et al., 2021; Du et al., 2017; Jacobacci et al., 2020), our results with the SRTT also provide evidence for a significant contribution of the micro-offline gains to overall learning in all four age groups. That is, performance improvements across blocks of training were largely due to micro-offline – as compared to micro-online – performance gains. However, and independent of the specific task variant, one cannot rule out the effect of fatigue on these results.”

c) Influence of task variants on changes in learning and memory across the lifespan

The reviewer recommended a substantial re-structuring of the introduction to better contextualize the influence of task variants on motor sequence learning and memory consolidation processes across the lifespan. However, it is our opinion that this change would not benefit the reader. A review of the literature fails to provide clear evidence that specific task variants lead to different outcomes in behaviors of interest. In the table below, we provide an overview of the literature presented in our introduction and specify the MSL task variant used.

Reported Results	Task Employed	
	SRTT	FTT
1. Assessment of differences in initial learning between children and adults	Du et al., 2017 Salehi et al., 2016 Meulemans et al., 1998 Wilhelm et al., 2013	Adi-Japha et al., 2014 Wilhelm et al., 2008 Ashtamker & Karni, 2013

	Janacsek et al., 2012 Lukács & Kemény, 2015 Thomas et al., 2004 Fischer et al., 2007 Juhasz et al., 2019 Wilhelm et al., 2012	
2. Children exhibit impaired sleep-related consolidation	Fischer et al., 2007 Wilhelm et al., 2013 Zinke et al., 2017	Wilhelm et al., 2008
3. Accelerated consolidation over wake intervals in children		Adi-Japha et al., 2014 Ashtamker & Karni, 2013 Dorfberger et al., 2007
4. Greater micro-offline learning in children	Du et al., 2017	
5. Comparable initial learning between young and older adults	Fitzroy et al., 2021 Spencer et al., 2007	Bottary et al., 2016 Fogel et al., 2014 Gudberg et al., 2015
6. Impaired consolidation in older adults	Brown et al., 2009 Spencer et al., 2007 Wilson et al., 2012	Bottary et al., 2016 Fogel et al., 2014 Gudberg et al., 2015

For most reported results shown, there are supporting citations for both the SRTT and finger tapping tasks (FTT). Accordingly, it is our opinion that re-writing the introduction to frame results in the context of specific MSL task variants would be of little value. There are a few examples in the table in which a reported result has been shown in only one of the two task variants (#3 and #4). Although it is our opinion that there is not enough literature to suggest that these results are indeed specific to a task variant, we do agree with the reviewer that it would be beneficial in these instances to explicitly state the task used (see lines 63-66, 56-59 and copied below for completeness).

“Specifically, Du et al. (2017) employed a stepping version of a SRTT and found that young children (i.e., 6-year-olds) showed higher micro-offline gains as compared to older children (i.e., 10-year-olds) and young adults.”

“Interestingly, there is some evidence from sequential finger tapping task variants suggesting that this degraded consolidation over sleep may be the by-product of enhanced or accelerated consolidation over the wake epochs shortly following initial learning (Adi-Japha et al., 2014; Ashtamker & Karni, 2013; Dorfberger et al., 2007).”

d) Rationale for the choice of the motor task

We agree with the reviewer that the manuscript would benefit from a clearer rationale for the choice of task employed. This information was added to the updated manuscript (lines 234-239 and copied below).

We thank the reviewer for this suggestion.

“The SRTT was chosen for this research, in part, because of the online data acquisition protocol. It was assumed that participants, and young children in particular, could more easily comprehend and follow the instructions for a SRTT variant (i.e., press the key that spatially corresponds to the visual stimulus). This in contrast to the sequential finger tapping task, where fingers are assigned numerical values and participants are then instructed to perform the sequence of finger movements that corresponds to the explicitly provided sequence of numbers.”

e) Macro-offline performance gains and task variants

We are not entirely sure we follow the reviewer’s point with respect to sleep-dependent consolidation and the choice of task. The initial evidence supporting the idea that explicit awareness is necessary for sleep-dependent consolidation comes from an explicit version of the SRTT adopted by (Robertson, 2004). This is very similar to the task used in our research and thus one could consider our task as sleep-dependent.

Comment 3: It is unclear whether each individual was assigned a unique sequence to learn, or whether the same sequence was used for all participants. The same question applies to the random sequence. Were there common segments (e.g., triplets) shared by these two sequences? The magnitude of sequence learning was measured by the difference in RT between the learned sequence and the random sequence. It is possible that adult participants acquired the entire sequence as one (especially when they were told there was a pattern before they started the task, so presumably they attempted to and could easily figure out the specific eight-element sequence at the end of learning), while children may learn just some short segments. This could render response speed faster for the common segments between the learned sequence and the random sequences in children, which possibly explains why children responded faster in the random sequences (figure 2A) and thus lower specific sequence learning magnitude. Clarification of these details is crucial for a comprehensive understanding of the study’s results.

Author response: We thank the reviewer for raising these issues. Our revised manuscript now includes additional information with respect to the structure of the sequential and pseudo-random task variants. This information is on lines 252-260 and copied below for completeness.

“In the sequential SRTT, participants were aware that the stimuli and keypresses followed a deterministic sequential pattern (i.e., 4-7-3-8-6-2-5-1, in which 1 through 8 are the left

pinky to the right pinky fingers from left to right), but they were not given any information about the structure or length of the sequence. All participants completed the same 8-element sequence. During the pseudorandom SRTT runs, the visual stimuli (and thus corresponding key presses) appeared in an order that pseudo-randomly changed every 8 elements. Specifically, the stimulus appeared in each location once every eight elements; and the stimulus never appeared in the same location consecutively. The order of the stimuli thus changed every eight elements within a block, across blocks of practice and differed across participants.”

As the location of the visual stimulus followed the pseudo-random structure described above, there were indeed triplets shared across sequential and random runs (i.e., triplets 4-7-3, 7-3-8, 3-8-6, etc. could appear in the pseudo-random run). However, the frequency of these shared triplets – averaged across age groups and the pre-learning and post-learning random runs - was very low (< 3%) and thus it is highly unlikely that shared *triplets* explain our results.

In deterministic motor sequence learning tasks such as the one employed in our research, researchers have often focused on transitions between keys (e.g., 4-7, 7-3, 3-8). Accordingly, we have decided to focus the rest of this response in the context of transitions as opposed to triplets. The frequency of shared transitions across groups and random runs was 14.21%, or equivalent to chance level (chance = 14.29% or 1 out of 7 given that keys were never repeated). As expected, the frequency of these shared transitions did not differ among groups (main effect of Group: $F_{(3,126)} = 0.153$, $p = 0.927$, partial $\eta^2 = 0.004$; run x group interaction: $F_{(3,126)} = 0.162$, $p = 0.922$, partial $\eta^2 = 0.004$). These data show that the number of sequence transitions that appeared in the pseudo-random runs was relatively small (approximately 1 out of 7) and did not differ among the 4 age groups.

The reviewer speculated that perhaps these shared transitions impacted RTs differentially across the 4 age groups. Notably, it was suggested that these shared transitions specifically led to faster (normalized) RTs in the post-learning random run in children, ultimately leading to smaller sequence-specific learning. To assess this possibility, we conducted a transition type (shared vs. unique) x group (4) mixed ANOVA on the non-normalized RT within the post-learning random run. [We focused on non-normalized data for this exploratory analysis because data normalization was done at the block – and not at the transition – level, the latter of which is necessary to address the reviewer’s comment.] Results showed the expected significant group effect ($F_{(3,126)} = 18.998$, $p < 0.001$, partial $\eta^2 = 0.311$), with no main effect of transition type ($F_{(1,126)} = 0.080$, $p = 0.777$, partial $\eta^2 = 0.001$). Interestingly, there was a significant transition type x group interaction ($F_{(3,126)} = 3.068$, $p = 0.030$, partial $\eta^2 = 0.068$). Follow-up pairwise comparisons assessing

differences between transition types within age groups (see Figure below) revealed faster RTs on the shared as compared to the unique transitions in young ($t_{(31)} = -1.945, p = 0.061$) and older adults ($t_{(31)} = -2.790, p = 0.009$), but not in the children ($t_{(32)} = 1.481, p = 0.148$) or adolescents ($t_{(32)} = -0.074, p = 0.941$). These data suggest that these shared transitions led to faster RTs in the post-learning random run in the two adult groups, but not in the children or the adolescents, which contrasts with the hypotheses put forth by the reviewer. Accordingly, the age group differences on the post-learning random were observed *despite* - as opposed to *because of* - these shared transitions. It is our opinion, however, that these additional analyses do not substantially contribute to the manuscript and thus we have elected to not include in the updated submission.

Average RTs during the post-learning random run for transitions that were shared (left plots) and unique (right plots) with the learned motor sequence. Shaded regions represent the kernel density estimate of the data, colored circles depict individual data, open circles represent group medians, and the horizontal lines depict group means (Bechtold et al., 2021).

Comment 4: A related comment is whether the specific sequence learning difference between children and adults is attributed to the explicit awareness of the existence of the sequence pattern. It has been reported that trying to figure out the pattern of a complex sequence could hinder sequence learning (Fletcher, Zafiris, Frith, Honey, Corlett, Zilles, Fink, 2004, Cerebral Cortex). Although the sequence used by the authors was relatively simple to adults, it could still pose a significant challenge for children if they tried to puzzle out the sequence order.

Each sequence consisted of only eight elements. This means that adult participants may become aware that once they respond with a certain finger, they would not need to press that finger for a few subsequent trials. This is different from traditional SRT tasks in which each stimulus appears twice in a single repetition

of a sequence. Do you think this might be the reason why adults demonstrated greater online learning especially that participants were told the existence of the sequence pattern? It would be helpful if the authors measured the explicit knowledge of the acquired sequence after participants finished the first session, but I think this might not be feasible because participants needed to perform the 5-hr and 24-hr retests.

Author response: We thank the reviewer for these comments. The experiments did *not* include a test of explicit awareness. Thus, we cannot rule out the possibility that results are due, in part, to age group differences in awareness. A statement on this limitation was added to the discussion of the revised text on lines 848-852 and copied below for completeness.

“Third, explicit awareness was not assessed at the end of any session or experiment. No explicit recognition or recall test was implemented due to the online nature of the experiment and the potential interference with the offline processes in Experiment 2. Nevertheless, the lack of such a test impedes our ability to draw any conclusions regarding age groups differences in explicit awareness and how such a difference may affect the results.”

The reviewer speculated that perhaps age group differences in awareness contributed to differences in “online learning.” We are not entirely sure whether the reviewer is referencing “micro-online learning” or “learning magnitude” here so we offer a response for both.

With respect to the learning magnitude results, this potential explanation does not appear to be possible. As outlined on lines 485 – 492, the age group difference in learning magnitude could be attributed to the random – as opposed to the sequence – condition. Specifically, children exhibited larger improvements on the random task from the pre-learning to post-learning random runs. Differences in explicit awareness of the sequence cannot explain differences in performance on the random task and thus the age-related differences in sequence-specific learning do not appear to be attributed to differences in explicit awareness.

If the reviewer’s statement references micro-online learning, we agree that the explanation put forth is indeed possible. The added paragraph on lines 848-852 (and copied above) addresses this limitation.

Comment 5: It is a bit hard to follow why RT needed to be normalized when it was used to quantify specific sequence learning. The difference in RT between the learned sequence and the random sequence at the end of session 1 already accounted for the general motor improvement.

Author response: Normalized performance measures were presented in the main text to account for age-related performance differences at baseline (see Figures 2 and 4). We then used these normalized measures in the computations of each index – including sequence specific learning - for consistency and simplicity (i.e., readers can visually “derive” the indices by looking at Figures 2 and 4). Nonetheless, we do agree with the reviewer that assessing age group differences on the same indices - but computed with non-normalized performance measures - would be of interest. The corresponding results were added to Appendices 3 and 4 of the Supporting Information. In brief, the findings of these additional analyses were identical to those in the main text and thus the specific computations did not influence our findings. As our initial computations were pre-registered, we have elected to keep these results in the main text of our manuscript. However, we do reference these additional computations and analyses on lines 374-377 and 397-398 of the main text.

Comment 6: The authors discussed the fatigue effect as the alternative possible explanation of the micro-offline gain. However, I did not quite follow why the finding that micro-offline gain is related to hippocampal activities rejects the fatigue explanation. These two cited studies used the finger sequencing task in adult populations. It is still very possible that fatigue accumulates when children keep performing the same SRT task. In addition, the online gains for all groups were negative (Figure 3C), indicating that RT became slower within each block, could this be explained by the fatigue effect? The authors should refrain from downplaying this fatigue explanation.

Author response: Our initial discussion was intended to convey that the fatigue explanation is incompatible with the growing body of neuroimaging data demonstrating that learning-related patterns of activity in the hippocampus are reactivated during the interleaved micro-offline epochs. That is, while fatigue may play a contributory role in the magnitude of micro-offline performance changes, neuroimaging evidence still suggests that micro-offline periods afford an opportunity for a rapid motor memory consolidation process. We do acknowledge, however, that our initial discussion lacked clarity and we have substantially revised this section. This new version can be found on lines 712 – 728 of the revised text (and copied below for completeness).

“Lastly, similar to previous studies that employed the self-initiated, sequential finger tapping task in young adults (Bönstrup et al., 2019; Buch et al., 2021; Du et al., 2017; Jacobacci et al., 2020), our results provide evidence for a significant contribution of the micro-offline gains to overall learning in all four age groups. That is, performance improvements across blocks of training were largely due to micro-offline – as compared to micro-online – performance gains. However, one cannot rule out the effect of fatigue on these results. Gupta and Rickard (2022, 2023) have argued that the magnitude of the micro-offline gains in young adults can be attributed to the build-up of fatigue or reactive inhibition as a function of task practice within blocks. Specifically, it was suggested that performance deteriorates across practice within a block due to reactive inhibition and thus micro-offline performance gains are inflated due to this performance variable that is independent of learning and memory processes. However, it is worth noting that this view cannot account for recent neuroimaging data showing reactivation of sequence learning-related patterns of hippocampal activity during the interspersed rest periods (Buch et al., 2021; Gann et al., 2023; Jacobacci et al., 2020). These neuroimaging data then suggest that micro-offline epochs afford an opportunity for a rapid memory consolidation process. Additional research is necessary to definitively conclude whether these micro-offline performance gains are reflective of a memory consolidation process or are simply attributed to reactive inhibition.”

Comment 7: In the statistical analyses section, it says that repeated measures ANOVA was used, while in the results section, mixed effect models were used.

Author response: There seems to be some confusion. Our analyses included group (4) by block (16 or 4, depending on contrast) mixed ANOVAs, as outlined in the statistical analyses section. In the results section, we outline the findings of those mixed ANOVAs and thus our manuscript does not include any linear mixed effect models.

Comment 8: Were the post hoc analyses corrected for multiple pairwise comparisons? It was not specified in the methods section.

Author response: The post hoc pairwise comparisons were performed using a Tukey test, which includes a correction for multiple comparisons. The test used was added to line 315 in the revised text.

Comment 9: The random blocks at the end of day 1 were not described in the Figure 1 legend.

Author response: We thank the reviewer for noticing this omission. This information has been added to the figure caption.

REVIEWER #2:

Comment 1: This is a study of motor sequence learning using the SRTT. The authors trained 4 groups of participants: children, adolescents, adults, and older adults. They were trained before and after a sleep interval, and also a shorter interval, and consolidation was examined. The main findings are a) consolidation happens both across the 'micro-interval' and b) the children showed more consolidation across both intervals, as well as across the micro rest epochs during training. The study addresses a valid question, since the literature has suggested that children do not show sleep dependent consolidation on this type of task. The authors suggest several possible explanations for the fact that their results are out of keeping with this - including a potential criticism of the design in past studies, which compared sleep to wake. The study is well written and very clear, with a careful description of methods and results. Overall, it seems clear that the study deserves publication. I do not find it especially exciting, yet I cannot find any errors or deficits that clearly need amendment either.

Author response: We thank the reviewer for the positive comments with respect to the rigor of our research.

REVIEWER #3:

Comment 1: The empirical study and the manuscript contain fascinating data. The issue of memory consolidation is rarely examined from a developmental perspective. Beyond the developmental curve, an important and exciting result is that ultra-fast consolidation can predict longer consolidation processes.

Author response: We thank the reviewer for acknowledging the positive contributions of this research.

Comment 2: The authors employed a visuomotor task in their studies. In these types of tasks, participants can acquire proficiency even when solely observing the screen without physically pressing any buttons. In light of the brief review by Pedraza et al. (2023) "Nomen est omen: Serial reaction time task is not a motor but a visuomotor learning task," European Journal of Neuroscience, Volume 58, Issue 4, on pages 3111-3115, it is suggested that the term "visuomotor learning" be incorporated into the title and manuscript.

Author response: We thank the reviewer for highlighting this short review. Reviewer 1 also raised some concerns with respect to specific task variants. Accordingly, we kindly refer this reviewer to read our responses to comment 2 from Reviewer 1 above for a more in-depth discussion on the matter. To directly speak to the reviewer's recommendation, the abstract now includes the phrase "Using a visually-cued serial reaction time task" to make this information clearer to the reader. Along the same lines, we added the word "visual" in the following sentence (lines 241-242):

"The SRTT employed in this research consisted of an 8-choice reaction time task in which participants were instructed to react to visual cues shown on a screen."

We have elected to not change our title for the following reasons. Our results are largely in line with previous research that employed the self-initiated (and thus not visually-cued) finger tapping paradigms (Adi-Japha et al., 2014; Ashtamker & Karni, 2013; Bottary et al., 2016; Dorfberger et al., 2007; Fogel et al., 2014; Gudberg et al., 2015). Accordingly, it is our opinion that the results appear to be generalizable across task variants and altering the title would be unnecessarily restrictive. We also choose not to place too much emphasis on the visual aspect in the title to avoid giving readers the impression that we employed an oculomotor sequence learning paradigm as done in previous research (Albouy et al., 2006, 2008; Gonzalez et al., 2016).

Comment 3: Prior to the authors' study, only one investigation had explored memory consolidation from a lifespan perspective: Tóth-Fáber et al. "Lifespan developmental invariance in memory consolidation: evidence from procedural memory," PNAS Nexus, Volume 2, Issue 3, in 2023, pgad037. While the authors correctly cited this study, it is worth noting that in the discussion section, they erroneously refer to it as

concerning initial learning, when in fact, it is a study on memory consolidation. Given that this study constitutes the sole prior research related to the authors' work, the introduction should be more closely aligned with the findings of this study. This adjustment does not diminish the significance of the authors' research, as it pertains to a different sequence type and contains numerous other compelling results. However, it can enhance the manuscript's overall coherence and relevance.

Author response: We apologize for the error in citing this paper. We have clarified this issue (see line 647). We have also added this citation in the introduction when referencing previous research that adopted a lifespan approach to learning and memory processes.

Although there is limited previous research that has employed a lifespan approach to motor learning and memory consolidation processes, we disagree that this one paper is the “sole prior research related to the authors’ work.” There are multiple papers that have examined learning and memory in children and older adults separately that are directly relevant to this work. These papers are incorporated into our introduction to provide a balanced perspective.

It is also worth noting that our section in the introduction with respect to the developmental differences in offline consolidation is centered around apparent differences when assessing consolidation over post-learning wake vs. sleep intervals. Specifically, there is evidence that children exhibit accelerated consolidation over post-learning wakefulness (Adi-Japha et al., 2014; Ashtamker & Karni, 2013; Dorfberger et al., 2007) but impaired sleep-dependent consolidation (e.g., Fischer et al., 2007; Wilhelm et al., 2008, 2013; Zinke et al., 2017). As the Tóth-Fáber paper employed a retest following a 24-hour interval that included both wakefulness and sleep, it is not possible to situate their results in this context. However, and as mentioned above, this paper is referenced both in the revised introduction and discussion.

Comment 4: One of the most thrilling findings is the ability of ultra-fast consolidation to serve as a predictor for longer-term consolidation. Within this context, it is pertinent to address three critical issues that warrant dedicated paragraphs in the discussion. Firstly, we should delve into the question of individual differences. An essential inquiry pertains to why a substantial number of experimental participants exhibit mini offline consolidation while others do not. The second issue that demands careful consideration is the choice between deterministic and probabilistic sequences. It is worth exploring how this selection may impact the results and their implications. Lastly, the third issue revolves around the duration of the mini-

consolidation phase. It is crucial to investigate whether the length of this mini-consolidation phase holds any significance and, if so, to what extent. These discussions are intertwined with two relevant articles: Quentin et al (2021) "Statistical learning occurs during practice while high-order rule learning during rest period." NPJ Science of learning and Szucs-Bencze et al. (2023). Manipulating the rapid consolidation periods in a learning task affects general skills more than statistical learning and changes the dynamics of learning. Eneuro, 10(2).

Author response: We thank the reviewer for raising these interesting points and for calling our attention to these papers. With respect to the last two points raised by the reviewer (i.e., deterministic vs. probabilistic and the length of the rest periods), we have expanded on this in our revised discussion (lines 695-702 and copied below).

"Of note, recent research in young adults has produced inconclusive results with respect to the impact of the rest period durations. In a probabilistic sequence task, there was no effect of duration (15s vs. 30s vs. self-selected) on micro-offline performance changes (Szücs-Bencze et al., 2023), whereas a temporal gradient (10s vs. 20s) was found in resistance to an interfering sequence in a deterministic motor sequence task (Bönstrup et al., 2020). In summary, future studies are needed to further examine micro-learning processes in children, with systematic examinations into certain methodological choices such as the age of the participants, the length of the rest epochs and the choice of sequence task (i.e., probabilistic vs. deterministic)."

With respect to the reviewer's first point, we certainly agree that adopting an individual differences approach may be of interest to some readers. We did conduct exploratory (non-pre-registered) regression analyses assessing the relationship between dependent measures of interest (i.e., micro-online and -offline gains, learning magnitude and 5-/24-hour macro-offline gains) and the following set of potential explanatory variables: circadian preference (self-report), time of testing, sleepiness (based on Stanford Sleepiness Scale; Maclean et al., 1992), vigilance (psychomotor vigilance task – PVT; Dinges & Powell, 1985; Experiment 2 only), as well as self-reported sleep duration and sleep quality. For each dependent measure / explanatory variable pair, we built two models (identical to our approach described on lines 425-432 in the main text). The first was a simple regression that examined the relationship independent of Age Group. The second was a multiple regression model that included Age Group as a dummy variable to assess whether the relationship between explanatory and dependent variables differed between young adults (reference group) and the other 3 Age groups. Here, we first focus on the output with micro-offline

performance gains, as this is the dependent measure specifically mentioned by the reviewer. No significant relationships were found between micro-offline performance gains and chronotype, time of testing, sleepiness, and sleep quality. The relationships between these variables also did not differ between young adults and any of the other 3 age groups. The only significant effect revealed was a positive relationship – collapsed across Age groups - between self-reported sleep duration the night prior to the experiment and micro-offline gains ($t=2.45$; $p = 0.016$). Note that the relationship between these two variables did not differ between young adults and the other 3 age groups. Although somewhat interesting, this significant relationship is to be expected given the Age group differences in sleep duration (see Appendix 1; sleep duration decreased with age) and in micro-offline performance gains (see Figure 3D). Accordingly, the significant relationship between sleep duration and micro-offline gains when collapsing across age groups simply reflects the age group differences already reported in the paper, as opposed to “true” individual differences. Given the highly explorative nature of these analyses, the large number of statistical tests necessary to probe these relationships and the lack of critical knowledge gained, it is our opinion that these analyses do not contribute to the paper. It is worth noting, however, that these data are publicly available without restriction. We encourage interested individuals to access these data and conduct additional relevant analyses.

Comment 5: Related to the previous: Another thought that crossed my mind is that it would be intriguing to examine the relationship between the online and offline consolidation indicators, as well as the micro offline and macro offline regressions. It would be fascinating to explore these group differences (children performing better offline and worse online) at the level of interindividual differences (those who excel in offline performance but struggle online).

Author response: For the assessment of the relationship between micro- and macro-offline performance changes, we kindly refer the reviewer to Section 4.2.4 of the main text as well Appendix 6 of the revised Supporting Information).

Note that assessing the relationship between micro-online and micro-offline performance gains is of little value given the inter-relatedness of the associated computations. Specifically, and as described on lines 390-395,

“...micro-online changes were computed as the difference between the averaged normalized RT for the correct keypresses of the first and last sequence repetitions within each block (i.e., first sequence block n – last sequence block n). For micro-offline changes,

the difference between the averaged normalized RT of the last sequence repetition of one block and the first sequence repetition of the subsequent block (i.e., last sequence block n – first sequence block $n+1$) was computed.”

Both measures are based on differences in RT on the same sequences but in opposing directions (online = first(n) – last(n); offline: last(n) – first($n+1$)). Accordingly, when averaging over blocks of practice, these two indices have an almost perfect negative relationship ($b = -0.987$, $p < 0.001$ in a simple regression model; see image below).

Micro-offline gains plotted as a function of the micro-online gains (data points are color coded by group). In panel A, the regression line fits data collapsed across the 4 age groups. Panels B-D compare young adults to children (B), adolescents (C) and older adults (D). Corresponding statistical results are presented in the main text.

Comment 6: The participant description section is not entirely clear to me. If I understand correctly, the authors were able to involve 224 people, but according to the sample size calculator, only 130 people's data were used in Experiment 1, and in Experiment 2, there were only 108 people in the end. Is there an overlap between the two samples? Were all 224 people measured in total? Did everyone complete the learning session here? And did some of the participants only take part in Experiment 2? So, authors could describe this a bit more clearly.

Author response: There is indeed an overlap in the two samples of people in Experiments 1 and 2. This information can be found on lines 160 - 164 of the updated manuscript (and copied below for completeness).

“Note that the procedures of Experiment 1 (assessing initial motor sequence learning) were identical to the first session of Experiment 2 (assessing macro-offline consolidation). Accordingly, data from a subset of early participants from Experiment 2 (n = 42; 16 children, 19 adolescents, 4 young adults, 3 older adults) were included in the analyses of Experiment 1. Thus, these participants were included in the analyses for both experiments.”

We acknowledge that we could have added all participants that completed the Experiment 2 protocol in the analyses of Experiment 1. We chose not to do this for two reasons. One, and as pointed out by the reviewer, this would not be consistent with our sample size calculation and our pre-registration. Second, this would have created extreme differences in the number of participants in each age group, leading to potential issues in statistical analyses.

Comment 7: I think it's problematic that the learning phase ends with a random block. These random blocks can totally change the results compared to a study without random blocks. The authors mentioned it as a limitation. Since it is a crucial point, I would elaborate more on this issue in the manuscript.

Author response: The inclusion of a random task variant towards or at the end of an initial learning session has been done extensively in previous literature assessing offline consolidation processes (Bottary et al., 2016; Brown et al., 2009; Robertson, 2004; Robertson et al., 2005; Spencer et al., 2006, 2007; Wilson et al., 2012). The general assumption is that since there is no sequence to learn in random task variants, this would then *not* introduce any interference. We are not aware of any paper that has explicitly compared memory consolidation processes with and without end-of-training random blocks. Thus, it remains an open question as to whether the end-of-training random blocks impact offline consolidation processes. Nonetheless, we do recognize that the inclusion of random blocks could have impacted the results. This is indeed the reason for discussing this issue on lines 842 - 846 in our limitations section (copied below for completeness). It is our opinion that this issue is adequately discussed in our manuscript.

“Second, and specific to experiment 2, each session ended with a pseudorandom run. The implementation of this random run is often used in sequence learning studies and has the benefit of assessing sequence specificity of performance improvements (Du et al., 2017; Lukács & Kemény, 2015; Salehi et al., 2016; Savion-Lemieux et al., 2009; Thomas & Nelson, 2001). However, we cannot rule out the possibility that the random runs at the end of each session interfered with the macro-offline processing of the sequential motor memory.”

Comment 8: The results of Experiment 2 say that the initial learning session results are similar to those in Experiment 1, meaning there's no difference in learning. However, there was a difference in learning in Experiment 1 (even though it may be more related to general skill than sequence-specific learning).

Author response: We thank the author for this comment. This sentence was intended to reference the analyses of learning dynamics (i.e., performance improvements across sequence blocks as assessed with a block x group ANOVA), in which no differences were observed. We have altered the wording on lines 551-554 to avoid confusion.

Comment 9: In children, sequence-specific learning in the learning phase was influenced by motor learning (even though the performance was normalized to baseline RT), could this not play a role in macro consolidation? Although there is analysis for random macro-offline gain in the supplementary, there are in principle the same age groups.

Author response: We regret to state that we do not fully understand the reviewer's comment. Specifically, we do not follow the phrases "sequence-specific learning in the learning phase was influenced by motor learning" and "there are in principle the same age groups." Below we offer our best interpretation and a subsequent response. If this interpretation is incorrect, we are happy to address this comment further if the reviewer is willing to clarify.

We think that the reviewer is asking about the relationship between sequence-specific learning and macro-offline consolidation. To address this point, we performed regression analyses with the learning magnitude (i.e., sequence specific learning) of session 1 as an independent variable and the 5h or 24h sequential macro-offline gains as dependent variables in separate models. Subsequently, and to examine if these relationships differed among age groups, we included age group (dummy coded) as well as age group x learning magnitude interaction variables in the regression models. The young adult group was selected as the reference group and thus the interaction variables compare the regression slopes of the other 3 age groups with those of the young adults. Results from the simple regression (i.e., ignoring age group) revealed a significantly negative relationship between learning magnitude and both the 5h ($b = -0.168$; $p=0.028$) and 24h ($b = -0.259$; $p<0.001$) sequential offline gains. This indicates that a greater learning magnitude in session 1 was associated with smaller performance gains across the 5h and 24h offline periods following training. Importantly, this relationship was not significantly different between the young adults and any of the other age groups (all $p > 0.18$). Given the lack of differences among age

groups and the highly exploratory nature of these analyses, we have opted to not incorporate these results into the revised manuscript.

Comment 10: The study has three 7-year-olds in Experiment 1 and one in Experiment 2. The authors mentioned it in the limitations. But for me, it's a bit questionable whether it would have been better to exclude them, as such a small number of participants doesn't really represent 7-year-olds well.

Author response: We believe there is no justifiable reason to exclude the 7-year-old participants. The primary analyses focus on age *group* comparisons (i.e., 7-12-year-old children were grouped together) and thus do not provide specific conclusions about 7-year-olds. The manuscript does include more fine-grained analyses of age-related changes with age input as a continuous variable. These exploratory analyses are simply used to depict developmental trajectories and also are not used to make conclusions with respect to a specific age (e.g., 7-year-olds).

Comment 11: The Sequence-specific learning was double-normalized because first, all blocks were plotted against the baseline data, and then when the learning index was calculated (random minus sequence)/random, the data were again adjusted to the random. I find it particularly strange that the results for the learning indicator are not repeated in the case of non-standardized RTs. Additionally, their normalization procedure is intriguing to me, and I haven't come across it being used before. But this may be good. Question: how much does this affect the results? Maybe this learning indicator should not be divided by the baseline? It would be nice to look at it anyway and compare it to the double normalized one.

Author response: We thank the reviewer for bringing up this point. We kindly refer the reviewer to our response to comment 5 from Reviewer 1 who raised a similar point. In brief, the same results are observed regardless of computation (see Appendix 3 in the revised Supporting Information).

Comment 12: Related to the previous: the authors say that the raw data results are included in the supplementary, but as far as I can see, they only report average RT/ACC results per period, so separate for sequence and separate for random blocks - not interesting in my opinion and unnecessary to justify that children are slower at each stage. What would be more interesting/useful would be how the learning indicators change on the raw data compared to the standardized data. So, general skill/seq learning and micro online/offline consolidation should be shown on raw data (if for no other reason than to be

comparable with other research). In Appendix 5, the analyses in the supplementary provide some answers to the sequence-specific offline vs general skill offline problem, but not completely.

Author response: We thank the reviewer for this comment. We agree that it may be helpful to some readers to examine the dependent measures based on the non-normalized data. Results on the sequence-specific and micro-learning measures based on the non-normalized data in Experiment 1 were added to Appendix 3. Findings regarding the macro-offline performance changes based on the non-normalized data in Experiment 2 were added to Appendix 4. In brief, results stemming from the analyses of non-normalized data are consistent with those in the main text on normalized data. The one exception is the findings on the micro-learning measures. Specifically, the previously non-significant group effects reached significance (micro-online: $F_{(3,124)} = 8.280$, $p < 0.001$, partial $\eta^2 = 0.167$; micro-offline: $F_{(3,124)} = 8.499$, $p < 0.001$, partial $\eta^2 = 0.171$), with children showing significantly smaller micro-online and larger micro-offline performance gains as compared to all other age groups. Additionally, the block x group interaction effect on the micro-online gains also became significant ($F_{(30.005,1240.225)} = 1.553$, $p = 0.029$, partial $\eta^2 = 0.036$). Note that we have elected to keep these additional analyses in the Appendix whereas results corresponding to these various indices computed with the normalized RT have remained in the main text. This choice was made to stay consistent with our pre-registration. But we do reference these findings on lines 679-682 of the revised main text (and copied below).

“It is worth noting here that the assessment of micro-offline gains on non-normalized RT data did reveal significant age group effects, with children exhibiting larger gains than the other 3 age groups (see Appendix 3). This analysis, however, was not part of our pre-registration.”

Comment 13: Exp. 1, the micro-online and micro-offline indicators are not specific to seq-specific learning because they actually only compare block start vs. block end RT (micro-online) and block end vs. block start RT (micro-offline), which, of course, includes sequence, but also includes general skill learning.

Author response: The reviewer’s statement is indeed correct. This is true for all deterministic sequence learning tasks that do not interleave random trials within sequential blocks to avoid potential interference effects. It is worth emphasizing that the manuscript here does not claim the micro-learning measures to be sequence-specific and we employed analogous computations as done in previous research with similar tasks (Bönstrup et al., 2019, 2020; Gann et al., 2023; Jacobacci et al., 2020).

Comment 14: In Exp. 2, the "sequential offline changes" were only calculated on the sequence blocks, not combined/corrected with random blocks, so the same problem arises: it includes both sequence learning and general acceleration...

Comment 15: It would also be worth doing more precise analyses to be able to say more about sequence-specific learning, e.g., at the very least, authors should put the indicators calculated for seq and random blocks into the same ANOVA to see if there is an interaction between the two... Authors should discuss the Quentin's article (see above), because it dissects general skill and seq-specific, even if only for adults...

Author response: Please note that we grouped above two comments made by the reviewer, as they both fall under the general umbrella of the sequence-specificity of the macro-offline performance gains. The main text (and Figure 5) includes information on the macro-offline gains for the sequence condition. For offline gains in the random condition as well as sequence-specific gains, we kindly refer the reviewer to Appendix 8 for measures based on normalized data and Appendix 4 for measures based on non-normalized data. In brief, results revealed that the random macro-offline gains (i.e., offline performance changes on the post-test random runs) across both offline intervals were smaller in older adults than children and adolescents. Importantly, the children did not differ from any other age groups. The sequence-specific macro-offline gains (i.e., offline changes in the learning magnitude measure) across the two offline intervals were significantly larger in children as compared to the young and older adults. These results thus suggest that the offline changes in children involved the strengthening of their sequential memory and thus are indeed sequence-specific. This analysis is conceptually the same as putting random and sequence into the same ANOVA as suggested by the reviewer.

Comment 16: Authors interpret non-significant results as "similar performance", which in principle should not be allowed. Authors should carry out Bayesian analysis if they want to interpret non-significant results. Bayesian analysis warranted anyway.

Author response: We thank the reviewer for bringing up this important point. We have also updated the results reported in the main text to now include Bayes Factors (in addition to the already reported information on effect sizes and significance testing). This provides the reader with more information with respect to whether the observed data are more probable under the null model versus alternatives. The information that outlines the Bayesian approach employed in this manuscript is in lines 320-336. Bayes Factors are reported throughout the results section of the revised main text (see note below with respect to an impact of including these Bayes Factors on the interpretation of certain results). Last, we have also

adapted our wording to make it clear that our results do not provide evidence for an absence of an effect (e.g., see lines 475-476, 552-554, 644-645, 706-707, and copied below for completeness).

“And, we found no credible evidence that the four groups differed in these performance improvements.”

“We found little credible evidence for differences among age groups in the performance changes across training blocks.”

“On the one hand, our results provided no evidence that performance improvements across training differed among age groups, which was consistent with a subset of the previous literature.”

“Our results found no evidence for differences in micro-online and -offline performance gains between young and older adults.”

Interpretations from the reported Bayes Factors are predominantly in line with the interpretations of the corresponding null hypothesis statistical testing and thus our initial conclusions have remained largely unchanged. The one exception is with respect to the micro-online and -offline learning measures. The output of the null hypothesis statistical testing of the micro-online and -offline learning measures revealed trends for significant age group differences. Follow-up pairwise comparisons indicated significantly smaller and larger micro-online and offline performance gains, respectively, in children as compared to both young and older adults. However, none of the corresponding Bayes Factors (BFs) were large enough (according to criteria in Wagenmakers et al., 2011) to provide substantial evidence supporting the alternative hypothesis of age group differences. Given this information from the BFs, as well as only a trend for a significant difference in the omnibus one-way ANOVAs, we have elected to temper our interpretation of these data. We still report the full picture of the results outlined above, but we no longer offer the conclusion that children exhibited significantly smaller and larger micro-online and -offline performance gains. These updates can be found in the corresponding results section of the updated manuscript (lines 505-537). The text in the abstract, the discussion (lines 674-683) and conclusions (lines 877-879) have also been modified accordingly.

MINOR:

Comment 17: Line 23: n=238 datasets ?

Author response: The word ‘datasets’ was used to delineate from ‘participants’ given some participants were included in the analyses of both Experiments 1 and 2. Please see our response to comment 6 of Reviewer 3 for further explanation.

Comment 18: Line 182 Table 2 - I suggest to include the p values (from supplementary) to see the statistical comparison between the age groups.

Author response: We thank the reviewer for this suggestion. The p-values resulting from the one-way ANOVAs assessing group differences were added to Table 1 and 2 in the main text. The caption refers to Appendix 1 for full statistical information.

Comment 19: Line 567-570: It's not very clear what is presented here. This should be a Group x micro-Offline interaction, right? Where's the interaction p value?

Author response: This section presents the relationship between the offline changes in performance on the micro- and macro-offline timescales. Given that micro-offline gains were a continuous variable, we conducted a multiple linear regression with Age Group coded as a dummy variable (as opposed to a mixed ANOVA with a categorical variable). An explanation of these analyses can be found on lines 426 – 432 (and copied below):

“Specifically, we performed exploratory multiple regressions with the micro-offline gains as independent and the 5h or 24h macro-offline gains as dependent variables. To examine how these relationships differed among age groups, we included age group (dummy coded) main effects as well as age group x micro-offline gains interaction variables in the regression models. The young adult group was selected as the reference group and thus the interaction variables compare the regression slopes of the other 3 age groups with those of the young adults.”

The corresponding results are then beta estimates that represent the differences in slopes (which can be considered the group by micro-offline interaction) between the reference young adult group and each other age group. Accordingly, there are three betas comparing: a) young adults to children; b) young adult to adolescents; and c) young adults to older adults. These results are on lines 611 - 616 and 621-627 as well as in Appendix 6 (also copied below). Please note that some of the results changed due to the inclusion of the dummy variables in the regression models to account for the group differences in intercepts.

“Young adults exhibited a steeper slope as compared to children ($b = -1.371$, $p = 0.003$, $BF_{10} = 0.381$), adolescents ($b = -1.287$, $p = 0.002$, $BF_{10} = 0.476$) and older adults ($b = -1.509$, $p = 0.003$, $BF_{10} = 0.475$). However, the reported BFs indicate that the data are more consistent with no differences and these between-group differences were no longer statistically significant if the young adult participant with the extremely negative 5hr offline gain (see Figure 5a) was removed from the analyses.”

“This relationship was significantly different between the children and young adults ($b = -0.669$, $p = 0.047$, $BF_{10} = 6.268$). Specifically, whereas young adults exhibited a significant

positive relationship between micro- and macro-offline performance gains ($b=0.647$, $p = 0.007$, $BF_{10} = 6.380$), children did not ($b = -0.023$, $p = 0.90$, $BF_{10}=0.360$). The relationship between micro-offline and 24-hr macro-offline gains did not differ between young adults and the other two age groups (adolescents: $b = 0.126$, $p = 0.679$, $BF_{10} = 0.594$; older adults: $b = -0.217$, $p = 0.556$, $BF_{10} = 0.189$).

Appendix 6 - *“In brief, there was a significant positive relationship between micro-offline and 24hr macro-offline gains collapsed across groups. This relationship was significantly different between young adults and children (Panel B) but not between young adults and adolescents (Panel C) or older adults (Panel D).”*

Response Document References

- Adi-Japha, E., Badir, R., Dorfberger, S., & Karni, A. (2014). A matter of time: Rapid motor memory stabilization in childhood. *Developmental Science*, *17*(3), 424–433. <https://doi.org/10.1111/desc.12132>
- Albouy, G., King, B. R., Maquet, P., & Doyon, J. (2013). Hippocampus and striatum: Dynamics and interaction during acquisition and sleep-related motor sequence memory consolidation. *Hippocampus*, *23*(11), 985–1004. <https://doi.org/10.1002/hipo.22183>
- Albouy, G., Ruby, P., Phillips, C., Luxen, A., Peigneux, P., & Maquet, P. (2006). Implicit oculomotor sequence learning in humans: Time course of offline processing. *Brain Research*, *1090*(1), 163–171. <https://doi.org/10.1016/j.brainres.2006.03.076>
- Albouy, G., Sterpenich, V., Balteau, E., Vandewalle, G., Desseilles, M., Dang-Vu, T., Darsaud, A., Ruby, P., Luppi, P.-H., Degueldre, C., Peigneux, P., Luxen, A., & Maquet, P. (2008). Both the Hippocampus and Striatum Are Involved in Consolidation of Motor Sequence Memory. *Neuron*, *58*(2), 261–272. <https://doi.org/10.1016/j.neuron.2008.02.008>
- Ashtamker, L., & Karni, A. (2013). Motor memory in childhood: Early expression of consolidation phase gains. *Neurobiology of Learning and Memory*, *106*, 26–30. <https://doi.org/10.1016/j.nlm.2013.07.003>
- Bechtold, B., Fletcher, P., Seamusholden, & Gorur-Shandilya, S. (2021). *bastibe/Violinplot-Matlab: A Good Starting Point* (v0.1) [Computer software]. Zenodo. <https://doi.org/10.5281/ZENODO.4559847>
- Bönstrup, M., Iturrate, I., Hebart, M. N., Censor, N., & Cohen, L. G. (2020). Mechanisms of offline motor learning at a microscale of seconds in large-scale crowdsourced data. *Npj Science of Learning*, *5*(1), 7. <https://doi.org/10.1038/s41539-020-0066-9>
- Bönstrup, M., Iturrate, I., Thompson, R., Cruciani, G., Censor, N., & Cohen, L. G. (2019). A Rapid Form of Offline Consolidation in Skill Learning. *Current Biology*, *29*(8), 1346-1351.e4. <https://doi.org/10.1016/j.cub.2019.02.049>
- Bottary, R., Sonni, A., Wright, D., & Spencer, R. M. C. (2016). Insufficient chunk concatenation may underlie changes in sleep-dependent consolidation of motor sequence learning in older adults. *Learning & Memory*, *23*(9), 455–459. <https://doi.org/10.1101/lm.043042.116>
- Brown, R. M., Robertson, E. M., & Press, D. Z. (2009). Sequence Skill Acquisition and Off-Line Learning in Normal Aging. *PLoS ONE*, *4*(8), e6683. <https://doi.org/10.1371/journal.pone.0006683>
- Buch, E. R., Claudino, L., Quentin, R., Bönstrup, M., & Cohen, L. G. (2021). Consolidation of human skill linked to waking hippocampo-neocortical replay. *Cell Reports*, *35*(10), 109193. <https://doi.org/10.1016/j.celrep.2021.109193>
- Dinges, D. F., & Powell, J. W. (1985). Microcomputer analyses of performance on a portable, simple visual RT task during sustained operations. *Behavior Research Methods, Instruments, & Computers*, *17*(6), 652–655. <https://doi.org/10.3758/BF03200977>
- Dorfberger, S., Adi-Japha, E., & Karni, A. (2007). Reduced Susceptibility to Interference in the Consolidation of Motor Memory before Adolescence. *PLoS ONE*, *2*(2), e240. <https://doi.org/10.1371/journal.pone.0000240>
- Doyon, J., Gabbitov, E., Vahdat, S., Lungu, O., & Boutin, A. (2018). Current issues related to motor sequence learning in humans. *Current Opinion in Behavioral Sciences*, *20*, 89–97. <https://doi.org/10.1016/j.cobeha.2017.11.012>
- Du, Y., Valentini, N. C., Kim, M. J., Whittall, J., & Clark, J. E. (2017). Children and Adults Both Learn Motor Sequences Quickly, But Do So Differently. *Frontiers in Psychology*, *08*. <https://doi.org/10.3389/fpsyg.2017.00158>

- Fischer, S., Wilhelm, I., & Born, J. (2007). Developmental Differences in Sleep's Role for Implicit Off-line Learning: Comparing Children with Adults. *Journal of Cognitive Neuroscience*, *19*(2), 214–227. <https://doi.org/10.1162/jocn.2007.19.2.214>
- Fitzroy, A. B., Kainec, K. A., Seo, J., & Spencer, R. M. C. (2021). Encoding and consolidation of motor sequence learning in young and older adults. *Neurobiology of Learning and Memory*, *185*, 107508. <https://doi.org/10.1016/j.nlm.2021.107508>
- Fogel, S. M., Albouy, G., Vien, C., Popovicci, R., King, B. R., Hoge, R., Jbabdi, S., Benali, H., Karni, A., Maquet, P., Carrier, J., & Doyon, J. (2014). fMRI and sleep correlates of the age-related impairment in motor memory consolidation. *Human Brain Mapping*, *35*(8), 3625–3645. <https://doi.org/10.1002/hbm.22426>
- Gann, M. A., Dolfen, N., King, B. R., Robertson, E. M., & Albouy, G. (2023). Prefrontal stimulation as a tool to disrupt hippocampal and striatal reactivations underlying fast motor memory consolidation. *Brain Stimulation*, *16*(5), 1336–1345. <https://doi.org/10.1016/j.brs.2023.08.022>
- Gonzalez, C. C., Billington, J., & Burke, M. R. (2016). The involvement of the fronto-parietal brain network in oculomotor sequence learning using fMRI. *Neuropsychologia*, *87*, 1–11. <https://doi.org/10.1016/j.neuropsychologia.2016.04.021>
- Gudberg, C., Wulff, K., & Johansen-Berg, H. (2015). Sleep-dependent motor memory consolidation in older adults depends on task demands. *Neurobiology of Aging*, *36*(3), 1409–1416. <https://doi.org/10.1016/j.neurobiolaging.2014.12.014>
- Gupta, M. W., & Rickard, T. C. (2022). Dissipation of reactive inhibition is sufficient to explain post-rest improvements in motor sequence learning. *Npj Science of Learning*, *7*(1), 25. <https://doi.org/10.1038/s41539-022-00140-z>
- Gupta, M. W., & Rickard, T. C. (2023). *A Quantitative Model that Incorporates Reactive Inhibition but No Facilitating Consolidation Explains Single-Session Motor Sequence Practice Effects* [Preprint]. PsyArXiv. <https://doi.org/10.31234/osf.io/d5pv9>
- Jacobacci, F., Armony, J. L., Yeffal, A., Lerner, G., Amaro, E., Jovicich, J., Doyon, J., & Della-Maggiore, V. (2020). Rapid hippocampal plasticity supports motor sequence learning. *Proceedings of the National Academy of Sciences*, *117*(38), 23898–23903. <https://doi.org/10.1073/pnas.2009576117>
- Janacsek, K., Fiser, J., & Nemeth, D. (2012). The best time to acquire new skills: Age-related differences in implicit sequence learning across the human lifespan. *Developmental Science*, *15*(4), 496–505. <https://doi.org/10.1111/j.1467-7687.2012.01150.x>
- Jongbloed-Pereboom, M., Nijhuis-van Der Sanden, M. W. G., & Steenbergen, B. (2019). Explicit and implicit motor sequence learning in children and adults; the role of age and visual working memory. *Human Movement Science*, *64*, 1–11. <https://doi.org/10.1016/j.humov.2018.12.007>
- Juhász, D., Nemeth, D., & Janacsek, K. (2019). Is there more room to improve? The lifespan trajectory of procedural learning and its relationship to the between- and within-group differences in average response times. *PLOS ONE*, *14*(7), e0215116. <https://doi.org/10.1371/journal.pone.0215116>
- King, B. R., Hoedlmoser, K., Hirschauer, F., Dolfen, N., & Albouy, G. (2017). Sleeping on the motor engram: The multifaceted nature of sleep-related motor memory consolidation. *Neuroscience & Biobehavioral Reviews*, *80*, 1–22. <https://doi.org/10.1016/j.neubiorev.2017.04.026>
- Krakauer, J. W., Hadjiosif, A. M., Xu, J., Wong, A. L., & Haith, A. M. (2019). Motor Learning. In R. Terjung (Ed.), *Comprehensive Physiology* (1st ed., pp. 613–663). Wiley. <https://doi.org/10.1002/cphy.c170043>
- Lukács, Á., & Kemény, F. (2015). Development of Different Forms of Skill Learning Throughout the Lifespan. *Cognitive Science*, *39*(2), 383–404. <https://doi.org/10.1111/cogs.12143>
- Maclean, A. W., Fekken, G. C., Saskin, P., & Knowles, J. B. (1992). Psychometric evaluation of the Stanford Sleepiness Scale. *Journal of Sleep Research*, *1*(1), 35–39. <https://doi.org/10.1111/j.1365-2869.1992.tb00006.x>

- Meulemans, T., Van der Linden, M., & Perruchet, P. (1998). Implicit sequence learning in children. *Journal of Experimental Child Psychology*, *69*(3), 199–221. <https://doi.org/10.1006/jecp.1998.2442>
- Robertson, E. M. (2004). Skill Learning: Putting Procedural Consolidation in Context. *Current Biology*, *14*(24), R1061–R1063. <https://doi.org/10.1016/j.cub.2004.11.048>
- Robertson, E. M., Press, D. Z., & Pascual-Leone, A. (2005). Off-Line Learning and the Primary Motor Cortex. *The Journal of Neuroscience*, *25*(27), 6372–6378. <https://doi.org/10.1523/JNEUROSCI.1851-05.2005>
- Salehi, S. K., Sheikh, M., Hemayattalab, R., & Humaneyan, D. (2016). *The Effect of Different Ages levels and explicit-implicit Knowledge on Motor Sequence Learning*. *11*(18), 13157–13165.
- Savion-Lemieux, T., Bailey, J. A., & Penhune, V. B. (2009). Developmental contributions to motor sequence learning. *Experimental Brain Research*, *195*(2), 293–306. <https://doi.org/10.1007/s00221-009-1786-5>
- Spencer, R. M. C., Gouw, A. M., & Ivry, R. B. (2007). Age-related decline of sleep-dependent consolidation. *Learning & Memory*, *14*(7), 480–484. <https://doi.org/10.1101/lm.569407>
- Spencer, R. M. C., Sunm, M., & Ivry, R. B. (2006). Sleep-Dependent Consolidation of Contextual Learning. *Current Biology*, *16*(10), 1001–1005. <https://doi.org/10.1016/j.cub.2006.03.094>
- Szücs-Bencze, L., Fanuel, L., Szabó, N., Quentin, R., Nemeth, D., & Vékony, T. (2023). Manipulating the Rapid Consolidation Periods in a Learning Task Affects General Skills More than Statistical Learning and Changes the Dynamics of Learning. *Eneuro*, *10*(2), ENEURO.0228-22.2022. <https://doi.org/10.1523/ENEURO.0228-22.2022>
- Thomas, K. M., Hunt, R. H., Vizueta, N., Sommer, T., Durston, S., Yang, Y., & Worden, M. S. (2004). Evidence of developmental differences in implicit sequence learning: An fMRI study of children and adults. *Journal of Cognitive Neuroscience*, *16*(8), 1339–1351. <https://doi.org/10.1162/0898929042304688>
- Thomas, K. M., & Nelson, C. A. (2001). Serial Reaction Time Learning in Preschool- and School-Age Children. *Journal of Experimental Child Psychology*, *79*(4), 364–387. <https://doi.org/10.1006/jecp.2000.2613>
- Tóth-Fáber, E., Nemeth, D., & Janacsek, K. (2023). Lifespan developmental invariance in memory consolidation: Evidence from procedural memory. *PNAS Nexus*, *2*(3), pgad037. <https://doi.org/10.1093/pnasnexus/pgad037>
- Wagenmakers, E.-J., Wetzels, R., Borsboom, D., & Van Der Maas, H. L. J. (2011). Why psychologists must change the way they analyze their data: The case of psi: Comment on Bem (2011). *Journal of Personality and Social Psychology*, *100*(3), 426–432. <https://doi.org/10.1037/a0022790>
- Wilhelm, I., Diekelmann, S., & Born, J. (2008). Sleep in children improves memory performance on declarative but not procedural tasks. *Learning & Memory*, *15*(5), 373–377. <https://doi.org/10.1101/lm.803708>
- Wilhelm, I., Metzkw-Mészáros, M., Knapp, S., & Born, J. (2012). Sleep-dependent consolidation of procedural motor memories in children and adults: The pre-sleep level of performance matters. *Developmental Science*, *15*(4), 506–515. <https://doi.org/10.1111/j.1467-7687.2012.01146.x>
- Wilhelm, I., Rose, M., Imhof, K. I., Rasch, B., Büchel, C., & Born, J. (2013). The sleeping child outplays the adult's capacity to convert implicit into explicit knowledge. *Nature Neuroscience*, *16*(4), 391–393. <https://doi.org/10.1038/nn.3343>
- Wilson, J. K., Baran, B., Pace-Schott, E. F., Ivry, R. B., & Spencer, R. M. C. (2012). Sleep modulates word-pair learning but not motor sequence learning in healthy older adults. *Neurobiology of Aging*, *33*(5), 991–1000. <https://doi.org/10.1016/j.neurobiolaging.2011.06.029>
- Zinke, K., Wilhelm, I., Bayramoglu, M., Klein, S., & Born, J. (2017). Children's initial sleep-associated changes in motor skill are unrelated to long-term skill levels. *Developmental Science*, *20*(6), e12463. <https://doi.org/10.1111/desc.12463>

27th Feb 24

Dear Dr King,

Your manuscript titled "Children exhibit a developmental advantage in the offline processing of a learned motor sequence" has now been seen by 2 of the original reviewers, whose comments appear below. In light of their advice I am delighted to say that we are happy, in principle, to publish a suitably revised version in Communications Psychology under the open access CC BY license (Creative Commons Attribution v4.0 International License).

We therefore invite you to revise your paper one last time to address the remaining concerns of our reviewers and a list of editorial requests. At the same time we ask that you edit your manuscript to comply with our format requirements and to maximise the accessibility and therefore the impact of your work.

EDITORIAL REQUESTS:

SUBMISSION INFORMATION:

OPEN ACCESS:

Communications Psychology is a fully open access journal. Articles are made freely accessible on publication under a CC BY license (Creative Commons Attribution 4.0 International License). This license allows maximum dissemination and re-use of open access materials and is preferred by many research funding bodies.

For further information about article processing charges, open access funding, and advice and support from Nature Research, please visit <https://www.nature.com/commspsychol/article-processing-charges>

At acceptance, you will be provided with instructions for completing this CC BY license on behalf of all authors. This grants us the necessary permissions to publish your paper. Additionally, you will be asked to declare that all required third party permissions have been obtained, and to provide billing information in order to pay the article-processing charge (APC).

* TRANSPARENT PEER REVIEW: Communications Psychology uses a transparent peer review system.

On author request, confidential information and data can be removed from the published reviewer reports and rebuttal letters prior to publication. If you are concerned about the release of confidential data, please let us know specifically what information you would like to have removed. Please note that we cannot incorporate redactions for any other reasons.

* CODE AVAILABILITY: All Communications Psychology manuscripts must include a section titled "Code Availability" at the end of the methods section. We require that the custom analysis code supporting your conclusions is made available in a publicly accessible repository at this stage; please choose a repository that generates a digital object identifier (DOI) for the code; the link to the repository and the DOI must be included in the Code Availability statement. Publication as Supplementary Information will not suffice.

* DATA AVAILABILITY:

[link redacted]

Best regards,

Jennifer Bellingtier, and on behalf of Xiaoqing Hu

Jennifer Bellingtier, PhD
Senior Editor
Communications Psychology

Xiaoqing Hu, PhD
Editorial Board Member
Communications Psychology
orcid.org/0000-0001-8112-9700

REVIEWERS' EXPERTISE:

Reviewer #1 Motor memory

Reviewer #3 Motor memory, consolidation, development

REVIEWERS' COMMENTS:

Reviewer #1 (Remarks to the Author):

I sincerely thank the authors for their efforts in addressing all of my comments and questions. The paper looks great to me.

Line 716 seemed a word or two were missing.

Reviewer #3 (Remarks to the Author):

No further comments.